# LEARNING LATENT STRUCTURAL CAUSAL MODELS

## ABSTRACT

Causal learning has long concerned itself with the recovery of underlying causal mechanisms. Such causal modelling enables better explanations of out-of-distribution data. Prior works on causal learning assume that the causal variables are given. However, in machine learning tasks, one often operates on low-level data like image pixels or high-dimensional vectors. In such settings, the entire Structural Causal Model (SCM) – structure, parameters, *and* high-level causal variables – is latent and needs to be learnt from low-level data. We treat this problem as Bayesian inference of the latent SCM, given low-level data. We present BIOLS, a tractable approximate inference method which performs joint inference over the causal variables, structure and parameters of the latent SCM from known interventions. Experiments are performed on synthetic datasets and a causal benchmark image dataset to demonstrate the efficacy of our approach. We also demonstrate the ability of BIOLS to generate images from unseen interventional distributions.

## 1 INTRODUCTION

The ability to learn causal variables and the dependencies between them is a crucial skill for intelligent systems, which can enable systems to make informed predictions and reasoned decisions, even in scenarios that diverge substantially from those encountered in the training distribution (Schölkopf et al., 2021; Goyal & Bengio, 2022). In the context of causal inference, a Structural Causal Model (SCM) (Pearl, 2009) with a structure $\mathcal{G}$ and a set of mechanisms parameterized by $\Theta$, induces a joint distribution $p(Z_1, \ldots Z_d)$ over a set of causal variables. However, the appeal of SCM-based modeling lies in its capacity to represent a family of joint distributions, each indexed by specific interventions. Models can then be trained on samples from a subset of these joint distributions and can *generalize* to completely unseen joint distributions as a result of new interventions.

Existing work on causal discovery aims to infer the structure and mechanisms of SCMs from observed causal variables. Learning such a causal model can then be useful for a wide-variety of downstream tasks like generalizing to out-of-distribution data (Scherrer et al., 2022; Ke et al., 2021), estimating the effect of interventions (Pearl, 2009; Schölkopf et al., 2021), disentangling underlying factors of variation (Bengio et al., 2012; Wang & Jordan, 2021), and transfer learning (Schoelkopf et al., 2012; Bengio et al., 2019).

Figure 1: BN for prior works in causal discovery and structure learning

In high-dimensional problems typically studied in machine learning, neither the causal variables nor the causal structure relating them are known. Instead, the causal variables, structure and mechanisms have to be learned from high-dimensional observations, such as images (see Fig. 3). An application of interest is in the context of biology, where researchers are interested in understanding Gene Regulatory Networks (GRN). In such problems, the genes themselves are latent but can be intervened on, the results of which manifest as changes in the high-resolution images (Fay et al., 2023; Chandrasekaran et al., 2023). Here, the number of latent variables (genes) is known but the structure, mechanisms, and the image generating function remain to be uncovered. This serves as the motivation for our work. Particularly, we address the problem of inferring the latent SCM – including the causal variables $\mathcal{Z}$, structure $\mathcal{G}$ and parameters $\Theta$ – by learning a generative model of the observed high-dimensional data. Since these problems potentially have to be tackled in the low-data and/or non-identifiable regimes, we adopt a Bayesian formulation so as to model the epistemic

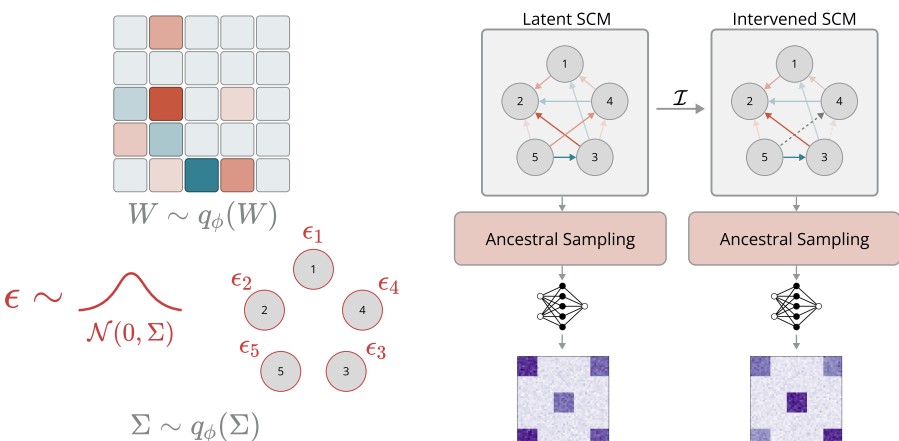

Figure 2: Architecture of BIOLS, our generative model for Bayesian latent causal discovery. BIOLS samples a weighted adjacency matrix ($W$) and covariance matrix ($\Sigma$) from posterior distributions. The matrix $\Sigma$ generates SCM noise variables ($\epsilon$). Together, $(W, \epsilon)$ define a latent SCM, generating latent causal variables and low-level data. The model supports latent SCM mutation and sampling from any interventional distribution.

uncertainty over latent SCMs. Concretely, given a dataset of high-dimensional observations, our approach BIOLS uses variational inference to model the joint posterior over the causal variables, structure and parameters of the latent SCM. Our contributions are as follows:

- We *propose a general algorithm, BIOLS, for Bayesian causal discovery in the latent space of a generative model*, learning a joint distribution over causal variables, structure and parameters in linear Gaussian latent SCMs with known interventions. Figure 2 illustrates an overview of the proposed method.

- By learning the structure and parameters of a latent SCM, we implicitly induce a joint distribution over the causal variables. Sampling from this distribution is equivalent to ancestral sampling through the latent SCM. As such, *we address a challenging, simultaneous optimization problem* that is often encountered during causal discovery in latent space: one cannot find the right graph without the right causal variables, and vice versa (Brehmer et al., 2022).

- On synthetically generated datasets and a benchmark image dataset (Ke et al., 2021) called the chemistry environment, BIOLS consistently outperforms baselines and uncovers causal variables, structure, and parameters. We also demonstrate the ability of BIOLS to generate images from unseen interventional distributions.

## 2 RELATED WORK

Prior work can be classified into Bayesian (Koivisto & Sood, 2004; Heckerman et al., 2006; Friedman & Koller, 2013) or maximum likelihood (Brouillard et al., 2020; Wei et al., 2020; Ng et al., 2022) methods, that learn the structure and parameters of SCMs using either score-based (Kass & Raftery, 1995; Barron et al., 1998; Heckerman et al., 1995) or constraint-based (Cheng et al., 2002; Lehmann & Romano, 2005) approaches.

**Causal discovery:** Within the realm of causal discovery, a significant body of work has emerged, primarily operating under the assumption that causal variables are directly observable and not derived from low-level data. Key contributions in this category include methods such as PC (Spirtes et al., 2000), Gadget (Viinikka et al., 2020), DAGnocurl (Yu et al., 2021), and Zhang et al. (2022). Notably, Peters & Bühlmann (2014) provides a foundational insight by proving the identifiability of linear Gaussian Structural Causal Models (SCMs) with equal noise variances. In the pursuit of causal discovery, various research directions have been explored. Bengio et al. (2019) employ the speed of adaptation as a signal for learning causal directions, while Ke et al. (2019) focuses on

learning causal models from unknown interventions. Further extending this exploration, Scherrer et al. (2021); Tigas et al. (2022); Agrawal et al. (2019); Toth et al. (2022) resort to active learning and for causal discovery, focusing on performing targeted interventions to efficiently learn the structure.Ke et al. (2022) proposed CSIvA, an approach that uses transformers (Vaswani et al., 2017) to learn causal structure from synthetic datasets and then generalize to more complex, naturalistic graphs. Ke et al. (2023) is a follow-up work to apply this to the problem of learning the structure of gene-regulatory networks. More recently, there have also been a host of methods using Generative Flow Networks (Bengio et al., 2021a;b) for causal discovery (Deleu et al., 2022; Nishikawa-Toomey et al., 2023; Deleu et al., 2023). Table 1 in the appendix compares BIOLS with prior work in Bayesian causal discovery.

Zheng et al. (2018) introduce an acyclicity constraint that penalizes cyclic graphs, thereby restricting search close to the DAG space. Lachapelle et al. (2019) leverages this constraint to learn DAGs in nonlinear SCMs. Building upon this constraint, Lachapelle et al. (2019) leverage it to learn DAGs within nonlinear SCMs. Temporal aspects of causal relationships are also explored, with methods like Pamfil et al. (2020) and Lippe et al. (2022) specializing in structure learning from temporal data.

**Causal representation learning**: Brehmer et al. (2022) present identifiability theory for learning causal representations from data pairs $(x, \tilde{x})$ before and after intervention on a single node, assuming fixed noise generated by the SCM. Ahuja et al. (2022) studies identifiability in a similar setup with the use of sparse perturbations. Ahuja et al. (2023) discusses identifiability for causal representation learning when one has access to interventional data. Kocaoglu et al. (2018); Shen et al. (2022); Moraffah et al. (2020) introduce generative models that use an SCM-based prior in latent space. In Shen et al. (2022), the goal is to learn causally disentangled variables. Yang et al. (2021) learns a DAG but assumes complete access to the causal variables. Given image labels and bounding boxes, Lopez-Paz et al. (2017) establishes observable causal footprints in images by trying to learn the causal direction between pairs of variables. Table 2 in the appendix situates BIOLS amidst related work in causal representation learning and generative causal models. In Appendix A, a more comprehensive exposition of prior works and their formulations is provided.

**Structure learning with latent variables**: Markham & Grosse-Wentrup (2020) introduce the concept of Measurement Dependence Inducing Latent Causal Models (MCM). The proposed algorithm seeks to identify a minimal-MCM that induces the dependencies between observed variables. Akin to VAEs, however, the method assumes the absence of causal links between latent variables. Kivva et al. (2021) delve into the conditions under which the number of latent variables and the underlying structure can be uniquely identified, specifically focusing on discrete latent variables, given the adjacency matrix between the hidden and measurement variables has linearly independent columns. Elidan et al. (2000) employs the concept of semi-cliques to detect the signature of hidden variables. Subsequently, they perform structure learning using the structural-EM algorithm (Friedman, 1998) for discrete random variables. Anandkumar et al. (2012) and Silva et al. (2006) tackle the identifiability of linear Bayesian Networks when certain variables are unobserved. However, it is important to note that the identifiability results in the former work are contingent upon specific structural constraints within the DAGs involved. Assuming non-Gaussian noise and that certain sets of latents have a lower bound on the number of pure measurement child variables, Xie et al. (2020) proposes the GIN condition to identify the structure between latent confounders. The frameworks discussed in the preceding works involve SCMs with a mix of observed and unobserved variables, whereas our setup considers the entirety of the SCM as latent. Lastly, GraphVAE (He et al., 2019) learns a structure between latent variables but does not incorporate notions of causality.

## 3 PRELIMINARIES

### 3.1 STRUCTURAL CAUSAL MODELS

A Structural Causal Model (SCM) is a framework that is primarily characterized by a set of equations that encapsulate how each endogenous variable, denoted as $Z_i$, is affected by its causal parents $Z^{\mathcal{G}}_{pa(i)}$ and an associated exogenous noise variable $\epsilon_i$. In this context, $Z^{\mathcal{G}}_{pa(i)}$ pertains to the collection of parent variables for $Z_i$. If the causal parent assignment is assumed to be acyclic, then an SCM is associated with a Directed Acyclic Graph (DAG) $\mathcal{G} = (V, E)$, where V corresponds to

the endogenous variables while the set $E$ encodes direct cause-effect relationships, signifying the underlying causal mechanisms.

The exact value $z_i$ assumed by a causal variable $Z_i$, is given by a local causal mechanism $f_i$, depending upon the values of its parent variables $z_{pa(i)}^{\mathcal{G}}$, a set of parameters $\Theta_i$, and the node-specific noise variable $\epsilon_i$. This relationship can be formally expressed through Equation 1. For the family of linear Gaussian additive noise SCMs, i.e. the setting that we focus on in this work, all $f_i$'s are linear functions, and $\Theta$ denotes the weighted adjacency matrix $W$, where each $W_{ji}$ is the edge weight of the connection $j \to i$. The linear Gaussian additive noise SCM thus reduces to equation 2.

$$z_i = f_i(z_{pa(i)}^{\mathcal{G}}, \Theta, \epsilon_i) \qquad (1) \qquad\qquad z_i = \sum_{j \in pa_{\mathcal{G}}(i)} W_{ji} \cdot z_j + \epsilon_i \qquad (2)$$

## 3.2 CAUSAL DISCOVERY

In prior research, the term "structure learning" typically denotes the process of learning a Directed Acyclic Graph (DAG) based on a specific optimization criterion, irrespective of whether causality is explicitly considered (e.g., as seen in GraphVAE He et al. (2019)). On the other hand, "causal discovery" is a more specialized endeavor aimed at uncovering both the structure and, in some instances, the parameters of SCMs, firmly grounded in the framework of causality and interventions (Pearl, 2009). Causal discovery methods seek to estimate the tuple $(\mathcal{G}, \Theta)$, where $\mathcal{G}$ is the DAG representing the causal structure, and $\Theta$ captures the parameters of the causal mechanisms.

These approaches often rely on modular likelihood scores that operate on the causal variables like the BGe score (Geiger & Heckerman, 1994; Kuipers et al., 2022) and the BDe score (Heckerman et al., 1995), to discern the right structure. Notably, these methods assume the availability of a dataset of observed causal variables. They typically aim to obtain a maximum likelihood estimate of either the structure alone or both the structure and parameters.

$$\mathcal{G}^* = \arg\max_{\mathcal{G}} p(Z \mid \mathcal{G}) \quad \text{or} \quad (\mathcal{G}^*, \Theta^*) = \arg\max_{\mathcal{G}, \Theta} p(Z \mid \mathcal{G}, \Theta), \qquad (3)$$

In the case of Bayesian causal discovery (Heckerman et al., 1997), variational inference is often employed to learn a joint posterior distribution $q_\phi(\mathcal{G}, \Theta)$ that approximates the true posterior $p(\mathcal{G}, \Theta \mid Z)$ by minimizing the KL divergence between them.

$$\min D_{\text{KL}}(q_\phi(\mathcal{G}, \Theta) \,\|\, p(\mathcal{G}, \Theta \mid Z)) \equiv \max \mathbb{E}_{(\mathcal{G}, \Theta) \sim q_\phi}\left[\log p(Z \mid \mathcal{G}, \Theta) - \log \frac{q_\phi(\mathcal{G}, \Theta)}{p(\mathcal{G}, \Theta)}\right], \qquad (4)$$

Here, $p(\mathcal{G}, \Theta)$ represents a prior distribution over the structure and parameters of the SCM, potentially encoding DAG-ness. Figure 1 illustrates the Bayesian Network (BN) used for inference in the context of causal discovery tasks.

## 3.3 LATENT CAUSAL DISCOVERY

In more practical scenarios, the learner is often unable to directly observe causal variables, necessitating their discovery from raw, high-dimensional data. In such cases, the causal variables, along with the underlying structure and parameters, form part of a latent Structural Causal Model (SCM). The primary objective of causal representation learning models is to facilitate both inference of, and generation from, the true latent SCM.

A notable work in this domain comes from Yang et al. (2021), that introduces a Causal Variational Autoencoder (Causal VAE). However, their approach operates within a supervised

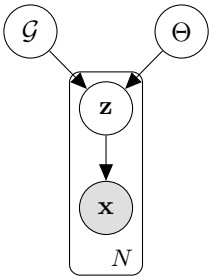

Figure 3: BN for the latent causal discovery task that generalizes standard causal discovery setups

framework where causal variables are labeled, with a specific emphasis on achieving disentanglement. In contrast, Kocaoglu et al. (2018) present causal generative models trained using adversarial techniques assuming observable causal variables. Their work, grounded in the presence of a known causal structure as a prior, focuses on generating data from conditional and interventional distributions.

In both the realms of causal representation learning and causal generative models discussed above, the Ground Truth (GT) causal graph and parameters of the latent SCM are arbitrarily defined on real datasets, and the setting is primarily supervised. In contrast, our setting operates in an unsupervised manner, where the primary goal is the recovery of the latent SCM responsible for generating the high-dimensional observed data. We term this problem "latent causal discovery". The BN for this setting is visually represented in Figure 3.

## 4 LEARNING LATENT SCMS FROM LOW-LEVEL DATA

### 4.1 PROBLEM SETUP

We are presented with a dataset $\mathcal{D} = \{\mathbf{x}^{(1)}, ..., \mathbf{x}^{(N)}\}$, where each $\mathbf{x}^{(i)}$ represents high-dimensional observed data. For simplicity, we assume that $\mathbf{x}^{(i)}$ is a vector in $\mathbb{R}^D$, but this setup can be extended to accommodate other types of inputs as well. Within this dataset, we posit the existence of latent causal variables $\mathbf{Z} = \{\mathbf{z}^{(i)} \in \mathbb{R}^d\}_{i=1}^N$, where $d \leq D$, which underlie and explain the observed data $\mathcal{D}$. These latent variables belong to a Ground Truth (GT) SCM, denoted by its structure $\mathcal{G}_{GT}$ and parameters $\Theta_{GT}$. We wish to invert the data generation process $g : (\mathcal{G}_{GT}, \Theta_{GT}) \to \mathbf{Z} \to \mathcal{D}$. In the setting, we also have access to the intervention targets $\mathcal{I} = \{\mathcal{I}^{(i)}\}_{i=1}^N$ where each $\mathcal{I}^{(i)} \in \{0, 1\}^d$. The $j^{\text{th}}$ dimension of $\mathcal{I}^{(i)}$ takes a value of 1 if node $j$ was intervened on in row entry $i$, and 0 otherwise. To formalize the setup, we consider $\mathcal{X}$, $\mathcal{Z}$, $\mathcal{G}$, and $\Theta$ to represent the random variables over low-level data, latent causal variables, the SCM structure, and SCM parameters, respectively.

### 4.2 BIOLS: BAYESIAN INFERENCE OVER LATENT SCMS

Here, we aim to estimate the joint posterior distribution $p(\mathcal{Z}, \mathcal{G}, \Theta \mid \mathcal{D})$ over the entire latent SCM. Computing the true posterior analytically requires calculating the marginal likelihood $p(\mathcal{D})$ which gets quickly intractable due to summation over the number of possible DAGs which grows super-exponentially with respect to the number of nodes. Thus, we resort to variational inference (Blei et al., 2017) that provides a tractable way to learn an approximate posterior $q_\phi(\mathcal{Z}, \mathcal{G}, \Theta)$ with variational parameters $\phi$, close to the true posterior $p(\mathcal{Z}, \mathcal{G}, \Theta \mid \mathcal{D})$ by maximizing the Evidence Lower Bound (ELBO),

$$\mathcal{L}(\psi, \phi) = \mathbb{E}_{q_\phi(\mathcal{Z}, \mathcal{G}, \Theta)} \left[ \log p_\psi(\mathcal{D} \mid \mathcal{Z}, \mathcal{G}, \Theta) - \log \frac{q_\phi(\mathcal{Z}, \mathcal{G}, \Theta)}{p(\mathcal{Z}, \mathcal{G}, \Theta)} \right], \quad (5)$$

where $p(\mathcal{Z}, \mathcal{G}, \Theta)$ is the prior distribution over the SCM, $p_\psi(\mathcal{D} \mid \mathcal{Z}, \mathcal{G}, \Theta)$ is the likelihood model with parameters $\psi$, mapping the latent causal variables to the observed high-dimensional data. An approach to learn this posterior could be to factorize it as

$$q_\phi(\mathcal{Z}, \mathcal{G}, \Theta) = q_\phi(\mathcal{Z}) \cdot q_\phi(\mathcal{G}, \Theta \mid \mathcal{Z}) \quad (6)$$

Given a procedure to estimate $q_\phi(\mathcal{Z})$, the conditional $q_\phi(\mathcal{G}, \Theta \mid \mathcal{Z})$ can be obtained using existing Bayesian structure learning methods (Cundy et al., 2021; Deleu et al., 2023). A high-level description is provided in section 3.2. Otherwise, one has to perform a hard simultaneous optimization which would require alternating optimizations on $\mathcal{Z}$ and on $(\mathcal{G}, \Theta)$ given an estimate of $\mathcal{Z}$. Difficulty of such an alternate optimization is highlighted in Brehmer et al. (2022).

**Alternate factorization of the posterior**: Rather than decomposing the joint distribution as in equation 6, we propose to only introduce a variational distribution $q_\phi(\mathcal{G}, \Theta)$ over structures and parameters, so that the approximation is given by $q_\phi(\mathcal{Z}, \mathcal{G}, \Theta) \approx p(\mathcal{Z} \mid \mathcal{G}, \Theta) \cdot q_\phi(\mathcal{G}, \Theta)$. The advantage of this factorization is that the true distribution $p(\mathcal{Z} \mid \mathcal{G}, \Theta)$ over $\mathcal{Z}$ is completely determined from the SCM given $(\mathcal{G}, \Theta)$ and exogenous noise variables (assumed to be Gaussian). This conveniently

avoids the hard simultaneous optimization problem mentioned above since optimizing for $q_\phi(\mathcal{Z})$ is avoided. Hence, equation 5 simplifies to:

$$\mathcal{L}(\psi, \phi) = \mathbb{E}_{q_\phi(\mathcal{Z}, \mathcal{G}, \Theta)} \left[ \log p_\psi(\mathcal{D} \mid \mathcal{Z}) - \log \frac{q_\phi(\mathcal{G}, \Theta)}{p(\mathcal{G}, \Theta)} - \cancel{\log \frac{p(\mathcal{Z} \mid \mathcal{G}, \Theta)}{p(\mathcal{Z} \mid \mathcal{G}, \Theta)}}^{0} \right] \tag{7}$$

where $p_\psi(\mathcal{D} \mid \mathcal{Z}, \mathcal{G}, \Theta) = p_\psi(\mathcal{D} \mid \mathcal{Z}) = \prod_{i=1}^{N} p_\psi(\mathbf{x}^{(i)} \mid \mathbf{z}^{(i)})$ according to the BN in figure 3 and is taken to be a Gaussian observation model. Such a posterior can be used to obtain an SCM by sampling $\hat{\mathcal{G}}$ and $\hat{\Theta}$ from the approximated posterior. As long as the samples $\hat{\mathcal{G}}$ are always acyclic, one can perform ancestral sampling through the SCM to obtain predictions of the causal variables $\hat{\mathbf{z}}^{(i)}$. For additive noise models like in equation 2, these samples are already reparameterized and differentiable with respect to their parameters. The predictions of causal variables are then fed to the likelihood model to predict samples $\hat{\mathbf{x}}^{(i)}$ that reconstruct the observed data $\mathbf{x}^{(i)}$.

### 4.3 Posterior parameterizations and priors

For linear Gaussian latent SCMs, which is the focus of this work, learning a posterior over $(\mathcal{G}, \Theta)$ is equivalent to learning $q_\phi(W, \Sigma)$, where $W$ refers to weighted adjacency matrices $W$ and $\Sigma$ refers to covariance of $p(\epsilon)$, the distribution over noise variables of the SCM with 0 means. Supposing $L$ to be the family of all adjacency matrices over a fixed node ordering, $W$ and $\Sigma$ parameterize the entire space of SCMs. Since $q_\phi(\mathcal{G}, \Theta) \equiv q_\phi(L, \Sigma)$, equation 7 leads to the following ELBO which has to be maximized, and the overall method is summarized in algorithm 1[1],

$$\mathcal{L}(\psi, \phi) = \mathbb{E}_{q_\phi(L, \Sigma)} \left[ \mathbb{E}_{q_\phi(\mathcal{Z} \mid L, \Sigma)} \left[ \log p_\psi(\mathcal{D} \mid \mathcal{Z}) \right] - \log \frac{q_\phi(L, \Sigma)}{p(L)p(\Sigma)} \right] \tag{9}$$

**Distribution over** $(L, \Sigma)$: The posterior distribution $q_\phi(L, \Sigma)$ has $\left( \frac{d(d+1)}{2} \right)$ elements to be learnt, and is parameterized by a diagonal covariance normal distribution. For the prior $p(L)$ over the edge weights, we promote sparse DAGs by using a horseshoe prior (Carvalho et al., 2009), similar to Cundy et al. (2021). A Gaussian prior is defined over $\log \Sigma$.

## 5 Experimental Results

In this section, we perform experiments to evaluate the learned posterior over the linear Gaussian latent SCM. We aim to highlight the performance of our proposed method on latent causal discovery. As proper evaluation in such a setting would require access to the GT causal graph that generated the high-dimensional observations, we test our method against baselines on synthetically generated vector data and also on a benchmark dataset called the chemistry environment (Ke et al., 2021) that causally generates images. Towards the end of the section, we evaluate the ability of our model to generate images from unseen interventional distributions.

**Baselines**: Since this is one of the early works to propose a working algorithm for this problem, we are not aware of baseline methods that solve this task. However, we compare our approach against 4 baselines – **VAE**, **GraphVAE**, **ILCM** and **ILCM-GT**. While VAE has a marginal independence assumption between latent variables, GraphVAE (He et al., 2019) learns a DAG structure over latent variables. The final two baselines are ILCM (as introduced in Brehmer et al. (2022)) and ILCM-GT, a variant of ILCM that directly uses the ground truth interventions instead of having to infer them. We include ILCM-GT to promote a fair comparison with BIOLS, since BIOLS does not infer interventions. It is to be noted that both VAE and GraphVAE are not designed to handle learning from interventional data. Additionally, while ILCM requires interventional data to train, the method also requires paired data inputs before and after an intervention with fixed noise over the unintervened nodes. For all baselines, we evaluate the quality of structure proposed by the learned model.

---

[1] The details for the ancestral sampling step can be found in appendix B

---

**Algorithm 1** Bayesian latent causal discovery to learn $\mathcal{G}, \Theta, \mathcal{Z}$ from high dimensional data

---

**Require:** $\mathcal{D}, \mathcal{I}$
**Ensure:** Approximate posterior distribution over $\mathcal{G}, \Theta, \mathcal{Z}$
 1: Initialize $q_\phi(L, \Sigma)$, $p_\psi(\mathcal{X} \mid \mathcal{Z})$, and set learning rate $\alpha$
 2: **for** num_epochs **do**
 3:     $(\widehat{L}, \widehat{\Sigma}) \sim q_\phi(L, \Sigma)$
 4:     $\widehat{W} \leftarrow \widehat{L}$
 5:     **for** $i \leftarrow 1$ to $N$ **do**
 6:         $\mathcal{C}^{(i)} \leftarrow \text{argwhere}(\mathcal{I}^{(i)} = 1)$
 7:         $\widetilde{W} = \text{copy}(\widehat{W})$
 8:         $\widetilde{W}[:, \mathcal{C}^{(i)}] \leftarrow 0$            ▷ Mutated weighted adjacency matrix according to $\mathcal{I}^{(i)}$
 9:         $\widehat{W}_{\mathcal{I}^{(i)}} \leftarrow \widetilde{W}$
10:         $\hat{\mathbf{z}}^{(i)} \leftarrow \text{AncestralSample}(\widehat{W}_{\mathcal{I}^{(i)}}, \widehat{\Sigma})$
11:     **end for**
12:     $\widehat{\mathbf{Z}} \leftarrow \{\hat{\mathbf{z}}^{(i)}\}_{i=1}^N$
13:     $\hat{\mathcal{D}} \sim p_\psi(\mathcal{X} \mid \mathcal{Z} = \widehat{\mathbf{Z}})$
14:     $\psi \leftarrow \psi + \alpha \cdot \nabla_\psi(\mathcal{L}(\psi, \phi))$            ▷ Update network parameters
15:     $\phi \leftarrow \phi + \alpha \cdot \nabla_\phi(\mathcal{L}(\psi, \phi))$
16: **end for**
17: **return** binary$(\widehat{W}), (\widehat{W}, \widehat{\Sigma}), \widehat{\mathbf{Z}}$

---

**Evaluation metrics**: We use two metrics commonly used in the literature – the expected Structural Hamming Distance ($\mathbb{E}$-**SHD**, lower is better) obtains the SHD (number of edge flips, removals, or additions) between the predicted and GT graph and then takes an expectation over SHDs of posterior DAG samples, and the Area Under the Receiver Operating Characteristic curve (**AUROC**, higher is better) where a score of 0.5 corresponds to a random DAG baseline. All our implementations are in JAX (Bradbury et al., 2018) and results are presented over 5 random DAGs.

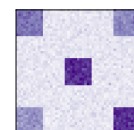

Figure 4: Image generated from the chemistry dataset.

**Generating the SCM**: Following many works in the literature, we sample random Erdős–Rényi (ER) DAGs (Erdos et al., 1960) with degrees in $\{1, 2\}$ to generate the DAG. For every edge in this DAG, we sample the magnitude of edge weights uniformly as $|L| \sim \mathcal{U}(0.5, 2.0)$. Each of the $d$ SCM noise variables is sampled as $\epsilon_i \sim \mathcal{N}(0, \sigma_i^2)$, where $\sigma_i \sim \mathcal{U}(1, e^2)$.

**Generating the causal variables and intervention targets**: We sample 20 random intervention sets where each set is a boolean vector denoting the intervention targets. An example of an intervention set for a 5-node DAG would be $[1, 0, 0, 1, 0]$. For each of these intervention sets, we generate 100 pairs of causal variables $(z_i, \tilde{z}_i)$ via ancestral sampling, where the intervention value is sampled from $\mathcal{N}(0, 2^2)$ with intervention noise added from $\mathcal{N}(0, 0.1^2)$. All interventions are taken to be hard and perfect. For nodes that were not intervened on, the same exogenous noise used to generate $z_i$ is used as in Brehmer et al. (2022). We now present 4 experimental setups to evaluate BIOLS where each experiment differs on how the generated causal variables are projected to $\mathcal{X}$.

$SO(n)$ **projection**: A random $SO(n)$ transformation is made on the generated causal variables to obtain pairs $(x, \tilde{x})$, as done in Brehmer et al. (2022).

**Linear projection**: A random projection matrix of shape $\mathbb{R}^{d \times D}$ is initialized with each entry of the matrix sampled from $\mathcal{U}(-5, 5)$. Following the linear projection, noise is sampled from $\mathcal{N}(\mathbf{0}, 0.1^2 * \mathbf{I_D})$, where $\mathbf{I_D}$ is the identity matrix of size $D$. In our experiments, we set $D$ to be 100.

**Nonlinear projection**: In more realistic scenarios, the ground truth generating function from $\mathcal{Z} \rightarrow \mathcal{X}$ is not just noisy but is also nonlinear. To this end, we initialize a random 3-layer neural network with ReLU activations to perform the projection from $d$ to $D$ dimensions. In this setting, $D$ is again set to 100. Noise is added while generating these observations in the same manner as in the previous projections.

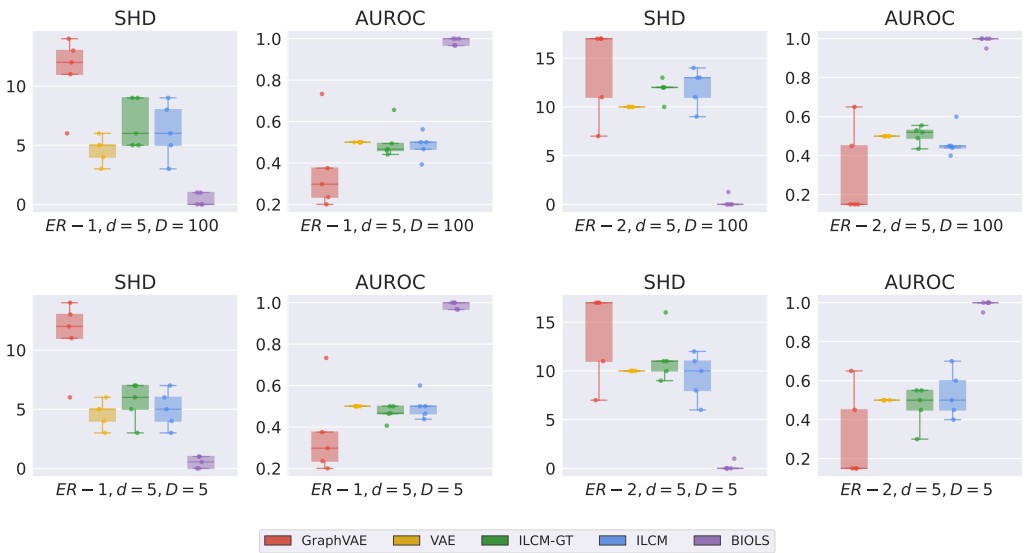

Figure 5: Learning $5-$node SCMs of different graph densities (ER-1 and ER-2) from a $100-$dimensional vector, where the generative function from $\mathcal{Z}$ to $\mathcal{X}$ is $SO(n)$ (row 1) and linear projection (row 2). $\mathbb{E}$-**SHD** ($\downarrow$), **AUROC** ($\uparrow$)

**Chemistry environment**: For this setting, intervention values are sampled from a standard Normal distribution instead. Once pairs of observational and interventional causal variables are generated, we use the chemistry environment to generate $(50, 50, 1)$ shaped images (Ke et al., 2021), wherein there are $d$ blocks with their intensity proportional to the $d$ causal variables. In order to maintain stochasticity in the ground truth image generation process, we add noise sampled from $\mathcal{N}(0, 0.05^2)$ to the normalized (from 0 to 1) pixels of the image and bring them back to the $0-255$ range.

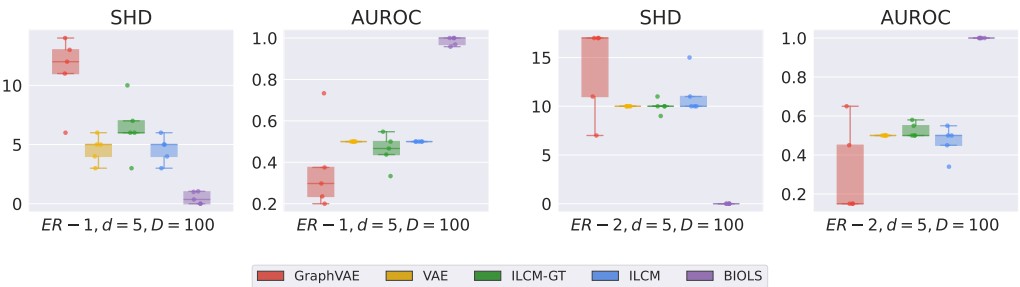

Figure 6: Learning $5-$node SCMs of different graph densities (ER-1 and ER-2) from a $100-$dimensional vector, where the generative function from $\mathcal{Z}$ to $\mathcal{X}$ is nonlinear. $\mathbb{E}$-**SHD** ($\downarrow$), **AUROC** ($\uparrow$)

In Figure 5, we present a summary of the results obtained from our proposed approach, BIOLS, for the first two experimental settings involving SCMs with 5-node DAGs, specifically focusing on $SO(n)$ and linear projection scenarios. Figure 6 further highlights the performance of BIOLS by showcasing the expected Structural Hamming Distance (SHD) and Area Under the Receiver Operating Characteristic (AUROC) for the learned models when the underlying generative function takes the form of an arbitrary nonlinear function.

We also perform experiments to evaluate latent causal discovery from image pixels and known interventions. The results are summarized in figure 7. It can be seen that BIOLS can recover the SCM significantly better than the baseline approaches on both metrics across all kinds of projections, including the challenging case of images. However, since running ILCM on image datasets was

not straightforward from the official implementation, we omit ILCM and ILCM-GT for the final experiment.

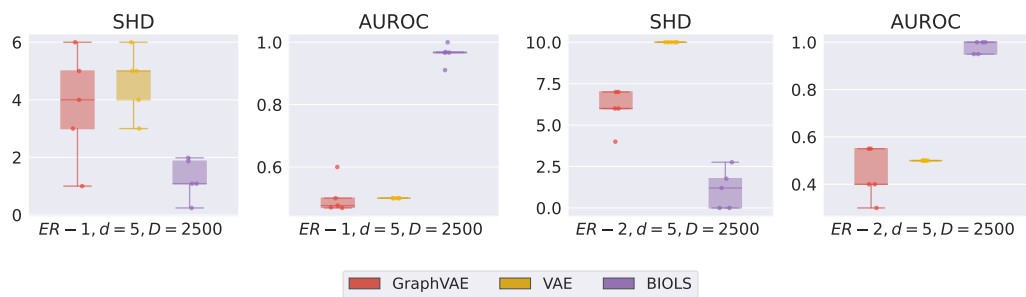

Figure 7: Learning $5-$node SCMs of different graph densities (ER-1 and ER-2) from $50 \times 50$ images in the chemistry benchmark dataset (Ke et al., 2021). $\mathbb{E}$-**SHD** ($\downarrow$), **AUROC** ($\uparrow$). Ground truth structure and weighted adjacency matrix for the ER-1 case is presented in figures 16 and 17.

In figure 8, we also evaluate the ability of the model to sample images from unseen interventional distributions in the chemistry dataset by examining the generated images with GT interventional samples. We notice that where there is a faint-colored (or a missing) block in the first row, there is a corresponding light-colored block in the second row. The matching intensity of each block corresponds to matching causal variables, demonstrating model generalization. In summary, *we note that BIOLS consistently outperforms baselines, while maintaining a low SHD between $0 - 1$.*

## 6 CONCLUSION

We presented a tractable approximate inference technique to perform Bayesian latent causal discovery that jointly infers the causal variables, structure and parameters of linear Gaussian latent SCMs under random, known interventions from low-level data. The learned causal model is also shown to exhibit generalization by sampling images from unseen interventional distribution. Our Bayesian formulation allows uncertainty quantification and mutual information estimation which is well-suited for extensions to active causal discovery. Extensions of the proposed method to learn nonlinear, non-Gaussian latent SCMs from unknown interventions would also open doors to general algorithms that can learn causal representations.

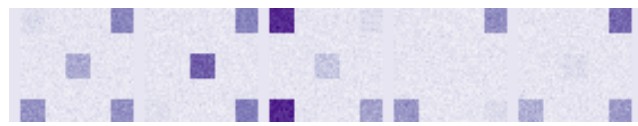

(a) Ground truth images sampled from 5 unseen interventional distributions.

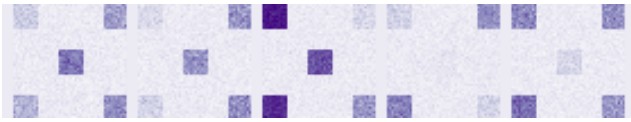

(b) Sampling images from the learned causal model, conditional on these 5 unseen interventions.

Figure 8: Samples of images from the ground truth and learned interventional distributions. Intensity of each block refers to the causal variable. One block is intervened in each column.

## 7 REPRODUCIBILITY STATEMENT

The code for all experiments, data generation, and the models used are all available at https://anonymous.4open.science/r/biols-3551.

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

## A   SITUATING BIOLS IN THE CONTEXT OF OTHER RELATED WORK

Table 1: Situating BIOLS in the context of related work in causal discovery.

| | Joint $G$ & $\theta$ | Unsupervised $Z$ | Nonlinear SCM | Learn from low-level data |
|---|---|---|---|---|
| VCN (Annadani et al., 2021) | ✗ | ✗ | ✗ | ✗ |
| DiBS (Lorch et al., 2021) | ✓ | ✗ | ✓ | ✗ |
| DAG-GFN (Deleu et al., 2022) | ✗ | ✗ | ✗ | ✗ |
| VBG (Nishikawa-Toomey et al., 2023) | ✓ | ✗ | ✗ | ✗ |
| JSP-GFN (Deleu et al., 2023) | ✓ | ✗ | ✓ | ✗ |
| **BIOLS (Ours)** | ✓ | ✓ | ✗ | ✓ |

In the causal representation learning section of related works (Section 2), we primarily cite six relevant works (Brehmer et al., 2022; Kocaoglu et al., 2018; Yang et al., 2021; Shen et al., 2022; Moraffah et al., 2020; Lopez-Paz et al., 2017), with Brehmer et al. (2022) serving as the primary baseline due to its high relevance to our explored setting (i.e., ILCM and ILCM-GT in section 5). Below, we provide a detailed discussion on each of the referenced works.

Brehmer et al. (2022) proves identifiability of latent causal models from low-level data under assumptions of paired low-level data $(x, \tilde{x})$ under random and unknown single target interventions on *every node* in the causal graph. In the setting, the number of causal variables are known apriori. Furthermore, for every sample of paired data, the noise is assumed to be fixed under the latent SCM. This is more of a counterfactual setting than an interventional one. In order to perform a fair comparison with Implicit Latent Causal Models (ILCM, the practical method proposed in Brehmer et al. (2022)), we stick to this setting of fixed noise between $x$ and $\tilde{x}$. However, BIOLS is broader in its applicability – it can handle *multi-target interventions*, scales to larger number of nodes, *does not require that the noise be fixed* before and after an intervention.

Kocaoglu et al. (2018) proposes causal GANs to learn observational and interventional image distributions, specifically for face-image datasets such as the labeled CelebA dataset(Liu et al., 2015). Notably, this work necessitates a causal graph over binary image label (e.g., young, smile, bald, narrow eyes), as illustrated in figure 5 of Kocaoglu et al. (2018). The work also assumes access to these labels and that the causal graph is already given. CausalVAE (Yang et al., 2021) is in a closely related setup and requires supervision, though it is not specific to face-image datasets. In contrast, our work operates beyond face-image datasets, does not necessitate image labels, is unsupervised, and centers around SCMs with *continuous variables*.

DEAR (Shen et al., 2022) addresses the training a causal disentangled generative model under supervision. However, this is done under the assumption that one has access to causal latent variables, and that the super-graph of the adjacency matrix is known before DAG learning. In comparison, BIOLS does not assume access to causal variables, making the setting distinct.

Causal Adversarial Networks (CAN) (Moraffah et al., 2020) is primarily designed for conditional or interventional image generation. Conditioning and interventions are on binary image labels (see figure 5 in Moraffah et al. (2020)), much akin to the setting of Kocaoglu et al. (2018). Notably, both CAN and CausalGAN concentrate on learning face-image distributions. However, the setting of CAN is not directly comparable to that of BIOLS. Adding CAN as a baseline is non-trivial and would introduce an apples-to-oranges comparison. BIOLS is distinct in its *focus on SCMs with continuous variables and does not involve labels*, unlike CAN.

In reference to Lopez-Paz et al. (2017), which explores causal inference in a bivariate setting, the study proposes training a binary classifier to identify plausible causal ($X \rightarrow Y$) and anticausal ($X \leftarrow Y$) relations with the aid of labels. However, it is crucial to note the distinctions between Lopez-Paz et al. (2017) and our work, BIOLS. The setup in Lopez-Paz et al. (2017) relies on images and assumes access to bounding boxes that highlight the presence of objects in the scene, and utilizes labels for the classification task. In contrast, BIOLS does not require bounding boxes or labels in its methodology. Furthermore, BIOLS can handle structure learning over multiple nodes, demonstrated in our experiments with up to $50$ nodes (refer section F).

Table 2: Situating BIOLS in the context of related work in causal generative models and causal representation learning.

| | Joint $G$ & $\theta$ | Learns any DAG | Unsupervised $Z$ | Scaling nodes | Cont. $Z$ | No constraints on paired data | Multi-target interventions |
|---|---|---|---|---|---|---|---|
| (Kocaoglu et al., 2018) | ✗ | ✗ | ✗ | ● | ✗ | ✓ | ● |
| (Yang et al., 2021) | ✓ | ✓ | ✗ | 4 | ● | ✓ | ● |
| (Shen et al., 2022) | ✗ | ✓ | ✗ | ● | ✓ | ✓ | ✗ |
| (Lopez-Paz et al., 2017) | ✗ | ✗ | ● | ● | ✓ | ✓ | ● |
| (Brehmer et al., 2022) | ✓ | ✓ | ✓ | $8-10$ | ✓ | ✗ | ✗ |
| **BIOLS (Ours)** | ✓ | ✓ | ✓ | 50+ | ✓ | ✓ | ✓ |

# B   IMPLEMENTATION DETAILS

$SO(n)$ **projection**: The construction of the projection matrix follows the methodology outlined in (Brehmer et al., 2022). However, for the sake of completeness, we provide a detailed description of the steps involved in obtaining this projection matrix. First, coefficients $c_{ij}$ are drawn from a Normal distribution, $c_{ij} \sim \mathcal{N}(0, 0.05^2)$, for every entry where $j < i$ in the lower triangle of a $\mathbb{R}^{d \times d}$ matrix. Subsequently, to ensure skew-symmetry, the upper triangular entries are populated with values such that $c_{ji} = -c_{ij}$. Finally, the matrix exponentiation process is applied to yield the desired projection matrix. This method ensures the matrix conforms to the special orthogonal group, $SO(n)$.

**Nonlinear projection**: A random 3-layer neural network with ReLU activations is initialized to execute the projection from $d$ to $D$ dimensions. While our reported experimental results focus on ReLU activations, it's important to note that BIOLS is versatile and supports nonlinear projections involving alternative activation functions, such as leaky ReLU, GeLU, among others. The network sizes for the 3-layer MLP are specified in Table 3. In all our experiments, $D$ is set to 100. For all our experiments, we use the AdaBelief (Zhuang et al., 2020) optimizer with $\epsilon = 10^{-8}$ and a learning rate of 0.0008. Our experiments are fairly robust with respect to hyperparameters and we did not perform hyperparameter tuning for any of our experiments. Table 4 summarizes the network details for the generative model $p_\psi(\mathcal{X} \mid \mathcal{Z})$.

Table 3: Network architecture for the nonlinear projection

| Layer type | Layer output | Activation |
|---|---|---|
| Linear | $D$ | GeLU |
| Linear | $D$ | GeLU |
| Linear | $D$ | |

Table 4: Network architecture for the decoder $p_\psi(\mathcal{X} \mid \mathcal{Z})$

| Layer type | Layer output | Activation |
|---|---|---|
| Linear | 16 | GeLU |
| Linear | 64 | GeLU |
| Linear | 64 | GeLU |
| Linear | 64 | GeLU |
| Linear | D | |

**Ancestral sampling from** $q_\phi(L, \Sigma)$: In section 4.3, we had mentioned that the posterior $q_\phi(L, \Sigma)$ is a $K$-variate Normal distribution with a diagonal covariance, where $K = \frac{d(d+1)}{2}$. This corresponds to a distribution over the $\frac{d(d-1)}{2}$ edges over the DAG (i.e, denoted by $L$, the lower triangular elements) and $d$ additional elements corresponding to exogenous noise variables $\epsilon = [\epsilon_1, \ldots, \epsilon_d]^T$ of the latent SCM (denoted by $\Sigma$). Suppose that the parameters of the Gaussian $q_\phi(L, \Sigma)$ is given by mean $\boldsymbol{\mu}_q$ and precision $\boldsymbol{P}_q$:

$$\boldsymbol{\mu}_q = (\boldsymbol{\mu}_L, \boldsymbol{\mu}_\Sigma) \qquad\qquad \boldsymbol{\mu}_L \in \mathbb{R}^{(K-d)} \quad \boldsymbol{\mu}_\Sigma \in \mathbb{R}^d \tag{8}$$

$$\boldsymbol{P}_q^{-1} = \begin{bmatrix} \boldsymbol{P}_L^{-1} & \mathbf{0} \\ \mathbf{0} & \boldsymbol{P}_\Sigma^{-1} \end{bmatrix} \qquad\qquad \boldsymbol{P}_L^{-1} \in \mathbb{R}^{(K-d) \times (K-d)} \quad \boldsymbol{P}_\Sigma^{-1} \in \mathbb{R}^{d \times d} \tag{9}$$

Consider $S \sim q_\phi(L, \Sigma)$. The first $K - d$ elements represent the weighted adjacency matrix $\widehat{W}$; for this, populate these $K - d$ elements in the lower triangle of a zero matrix. The last $d$ elements, $S_{(K-d):K}$, are samples of the predicted exogenous noise variables $\hat{\epsilon}$. Now, the SCM is defined by $\widehat{W}$ and $\hat{\epsilon}$, and ancestral sampling is expressed as $z_i := \widehat{W}_{*i}^T \mathbf{z} + \hat{\epsilon}_i$, where the ordering of assignment

indexed by $i$ is according to the topological ordering induced by $\widehat{W}$. $z_i$ is the $i^{\text{th}}$ element of $\mathbf{z}$, and $\mathbf{z}$ is initialized to zeros. $\widehat{W}_{*i}$ denotes the $i^{\text{th}}$ column vector of $\widehat{W}$.

## C ABLATION ON GRAPH DENSITY

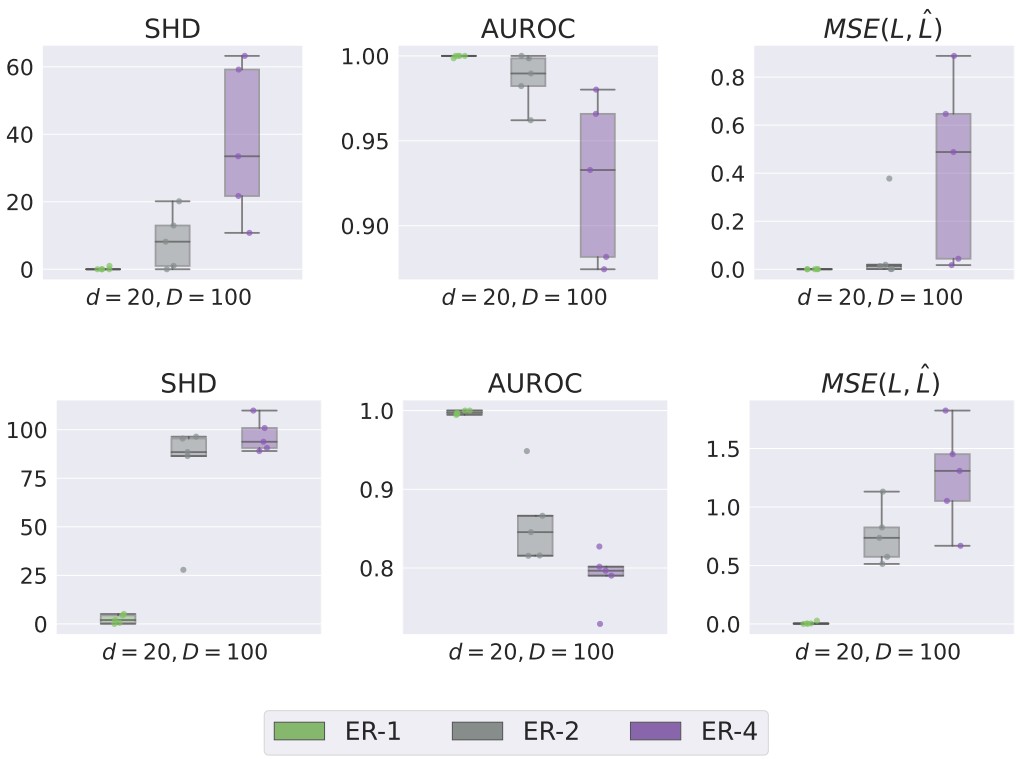

Figure 9: Effect of number of intervention sets on latent SCM recovery for linear (top row) and nonlinear (bottom row) generation function, $d = 20$ nodes. **SHD** $\downarrow$, **AUROC** $\uparrow$, **MSE**$(L, \hat{L})$ $\downarrow$

In this section, we perform ablations to study how graph density affects the quality of the SCM learned by BIOLS. Similar to other works in Bayesian structure learning, we notice that as the graph gets more dense, it gets harder to recover the SCM. Figure 9 illustrates the performance of BIOLS across the 3 metrics on $ER-1$, $ER-2$, $ER-4$ graphs. These studies are on $d = 20$ node DAGs, projected to $D = 100$ dimensions. The model is trained on 120 intervention sets, with 100 samples per set to stay consistent with rest of the experiments. The top row in the figure corresponds to a linear projection between latent and observed variables. Similarly, the bottom row corresponds to the nonlinear projection.

Notably, we observe a trend wherein recovering edges becomes more challenging with denser graphs. This difficulty may arise from BIOLS needing to uncover a greater number of cause-effect relationships. This observation aligns with insights often noted in traditional causal discovery algorithms (e.g., Fig 5 and 12 in Scherrer et al. (2021)). It is important to note that the performance for denser graphs can be further improved by providing more interventional data (see section D), increasing the variance of the Gaussian intervention values (refer section E), or both.

# D ABLATION ON NUMBER OF INTERVENTION SETS

In the data generation phase (section 5), it was observed that the interventional data is specified through two terms – number of intervention sets and number of interventional samples per set. Within this subsection, plots are presented to illustrate how the learning of the latent Structural Causal Model (SCM) is influenced for different numbers of nodes while the number of intervention sets is varied. The number of interventional samples per set remains constant and is fixed at 100, consistent with previous experiments. The impact of the number of intervention sets on the quality of the latent SCM recovered by BIOLS is depicted in figure 10 This analysis pertains to a linear projection to $D = 100$ for SCMs with 30 and 50 nodes. Similarly, Figure 11 presents a comparable plot for a nonlinear projection function.

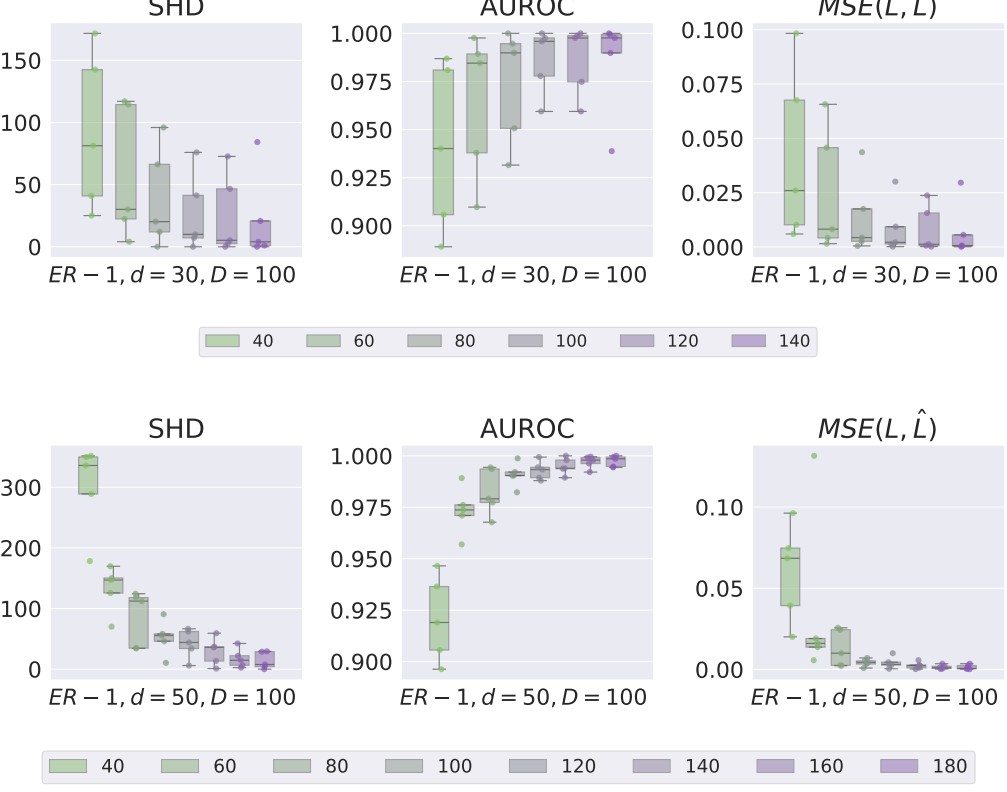

Figure 10: Effect of number of intervention sets on latent SCM recovery for a linear generation function, $d = 30, 50$ nodes. **SHD** $\downarrow$, **AUROC** $\uparrow$, **MSE**$(L, \hat{L})$ $\downarrow$

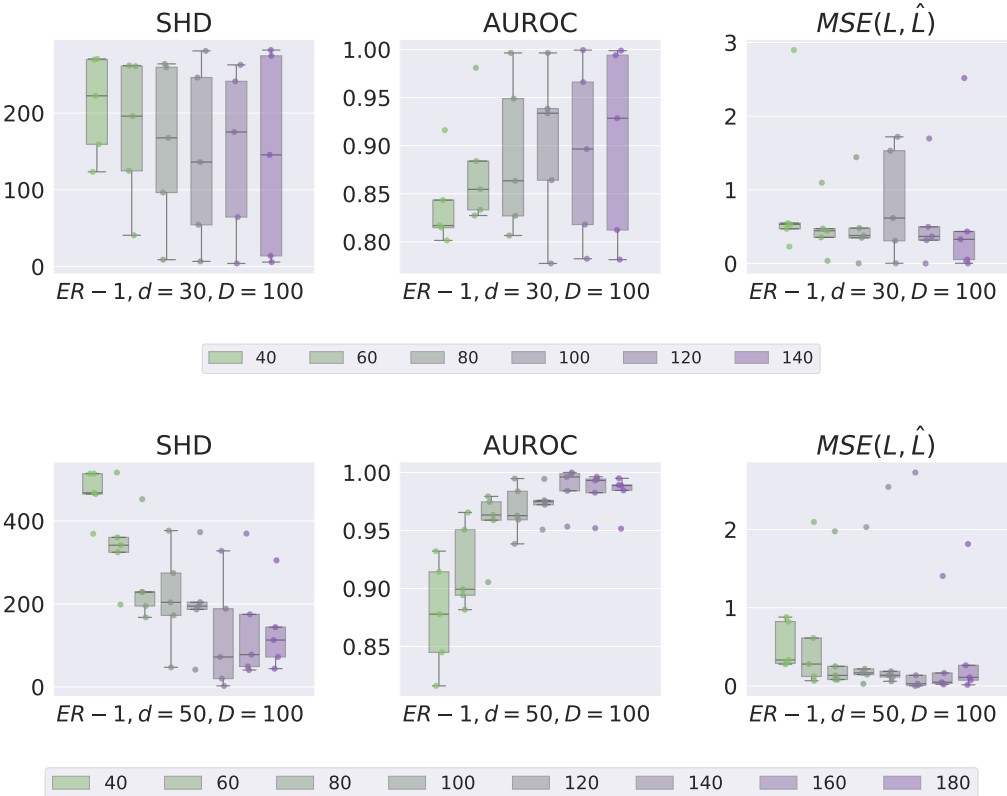

Figure 11: Effect of number of intervention sets on latent SCM recovery for a nonlinear generation function, $d = 30, 50$ nodes. **SHD** $\downarrow$ , **AUROC** $\uparrow$ , **MSE**$(L, \hat{L})$ $\downarrow$

# E ABLATION ON RANGE OF INTERVENTION VALUES

In many situations, interventions are useful, or even necessary, to learn about the causal structure. However, it is crucial to acknowledge that the performance of a causal discovery algorithm is also influenced by the *range of values* assigned to an intervened node. In general, one would anticipate the recovery of the SCM to be either comparable or improved when the intervention involves a broader range of values. To see why, consider a simple SCM: $A \rightarrow B$. Intervening on $A$ over a broad range of values, has a larger effect over values taken on by $B$.

Motivated by this example, we perform an ablation study to see if and how much the range of intervention values affect the performance of BIOLS. Figure 12 illustrates for a linear projection function, the performance of BIOLS for various number of intervention set where the intervention value is **(i)** deterministically set to 0 or **(ii)** sampled from a Normal distribution containing a larger range of values. Figure 13 illustrates a similar set of results but in the case of a nonlinear projection function. Similar to previous experiments, 100 interventional samples are collected for each intervention set.

We performed similar experiments where the intervention samples were drawn from $\mathcal{U}[-5, 5]$ and noticed similar results. That is, for the uniform intervention case, BIOLS performs significantly better compared to just using 0-valued interventions. However, for the sake of brevity, we do not provide these additional plots.

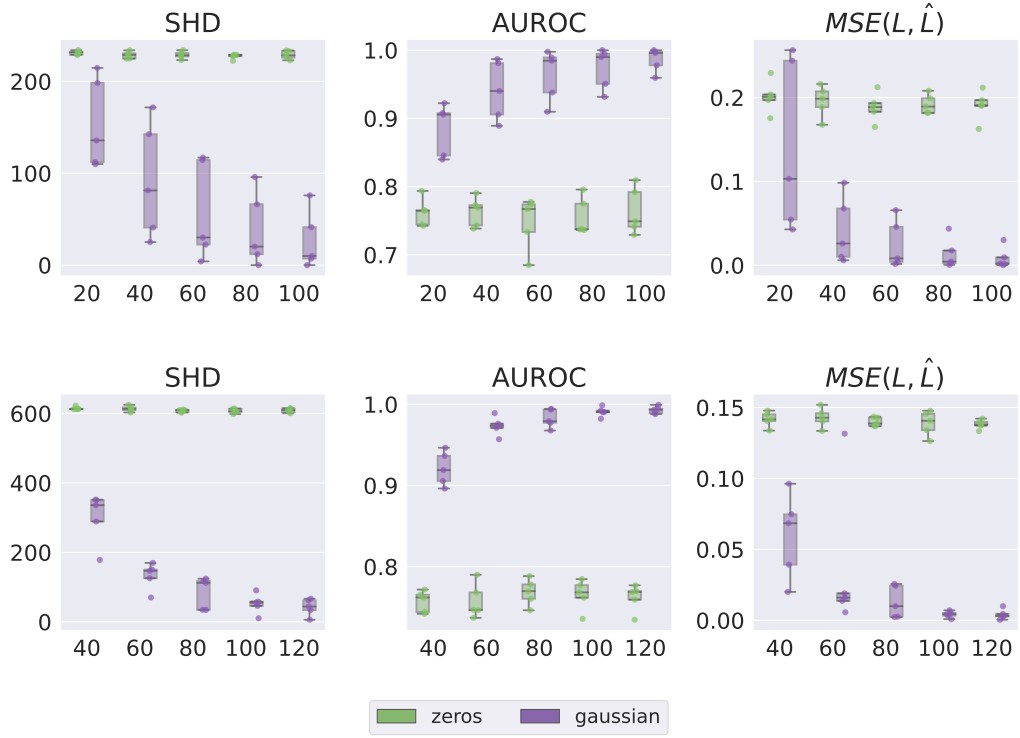

Figure 12: Effect of zero and stochastic (Gaussian) intervention values on latent SCM recovery, for a linear generation function, $d = 30, 50$ nodes. The X-axis refers to the number of intervention sets. **SHD** $\downarrow$ , **AUROC** $\uparrow$ , **MSE**$(L, \hat{L})$ $\downarrow$

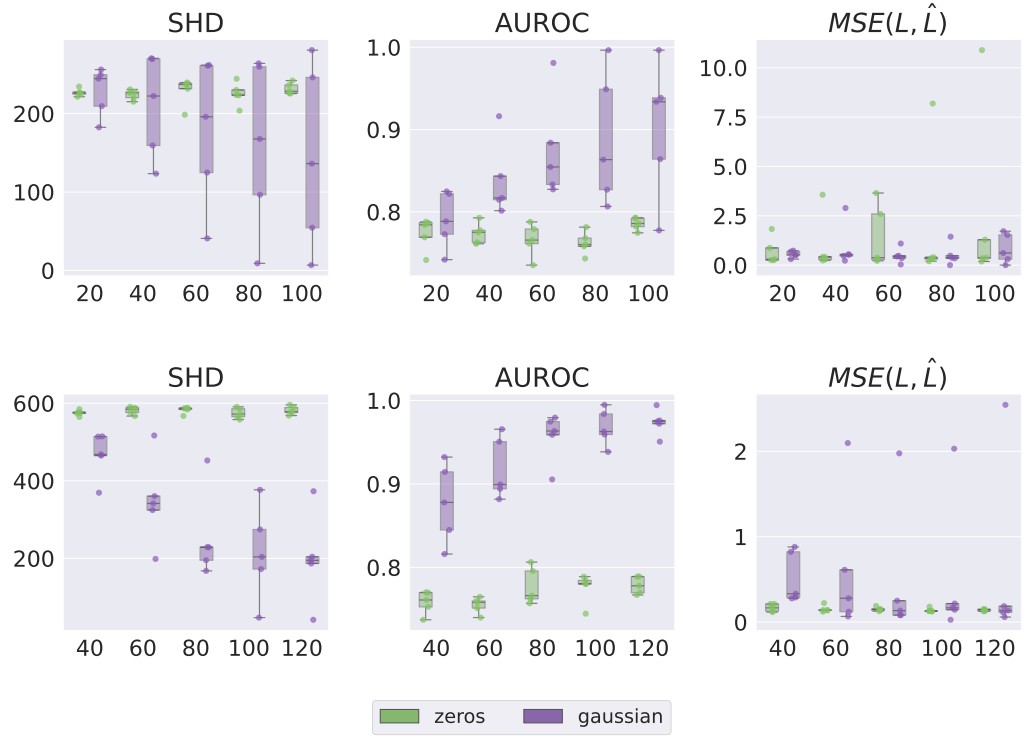

Figure 13: Effect of zero and stochastic (Gaussian) intervention values on latent SCM recovery, for nonlinear generation function modeled by a 3 layer MLP, $d = 30, 50$ nodes. The X-axis refers to the number of intervention sets. **SHD** $\downarrow$ , **AUROC** $\uparrow$ , **MSE**$(L, \hat{L})$ $\downarrow$

## F  SCALING THE NUMBER OF NODES

So far we have seen the performance of BIOLS in different settings such as learning graphs of varying densities (appendix C), learning from varying amounts of intervention data (appendix D), and learning from deterministic (zero-valued) interventions and stochastic interventions (appendix E). In this section, we study how BIOLS scales with the number of nodes in the SCM.

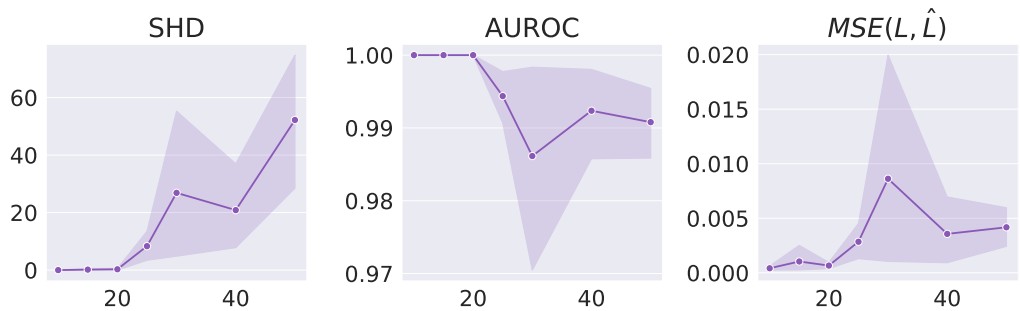

Figure 14: Scaling BIOLS across number of nodes for a linear data generation function, trained on 100 multi-target intervention sets with Gaussian intervention values.

Figure 14 plots the performance of BIOLS in a multi-target, Gaussian intervention setting. As usual, we use 100 interventional sets with 100 data points per intervention set, and evaluate across 5 seeds.

In figure 15, we summarize results for a similar experiment, where the projection function is nonlinear. The only difference in this experiment is that we use 400 interventional sets with 100 samples per set. We note that **BIOLS successfully scales to atleast upto** 50 **nodes**.

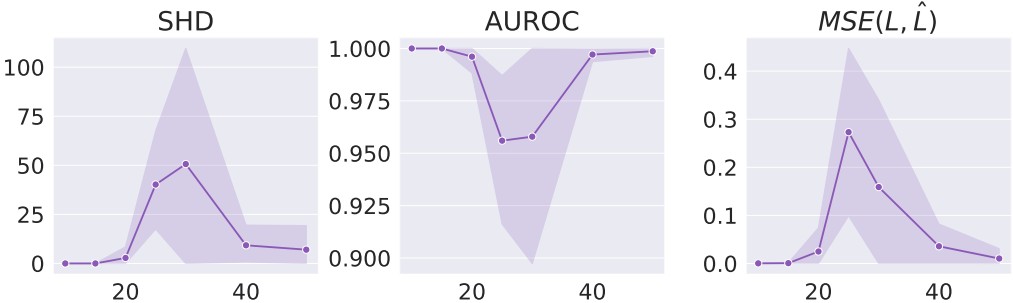

Figure 15: Scaling BIOLS across number of nodes for a nonlinear data generation function, trained on 400 multi-target intervention sets with Gaussian intervention values.

## G    RUNTIMES

Here, we note the runtimes for a subset of the previously presented experiments. We explore the program runtime along two axes: scaling with respect to nodes, and with respect to number of data points.

Table 5 provides the program runtime for scaling BIOLS, where $n$ corresponds to pairs of (observational, interventional) data, and $d$ denotes the number of nodes in the latent Structural Causal Model (SCM). All reported runtimes are based on 10,000 epochs of BIOLS across 5 seeds, with observed dimensions $D$ set to 100 and utilizing a linear projection function. It is noteworthy that the runtimes are consistent across experiments involving nonlinear projections. This consistency arises from the maintenance of identical neural network sizes, with the only variation being the data generation procedure.

|            | $n = 4000$ | $n = 6000$ | $n = 8000$ | $n = 10000$ | $n = 12000$ |
|------------|-----------|-----------|-----------|------------|------------|
| $d = 15$   | 12        | 15        | 17        | 20         | 23         |
| $d = 20$   | 20        | 24        | 26        | 29         | 35         |
| $d = 25$   | 28        | 32        | 36        | 40         | 45         |
| $d = 30$   | 40        | 47        | 50        | 55         | 60         |
| $d = 40$   | 72        | 76        | 90        | 100        | 105        |
| $d = 50$   | 105       | 116       | 127       | 142        | 166        |

Table 5: Program runtimes: Scaling BIOLS across number of nodes and data points, with $D = 100$. All runs are reported on 10000 epochs of BIOLS across 5 seeds. All runtimes are reported in minutes.

## H    ADDITIONAL VISUALIZATIONS

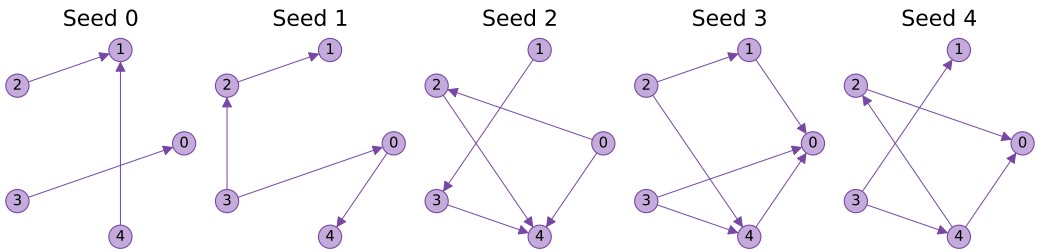

Figure 16: Ground truth causal structures for the experiment on the chemistry dataset.

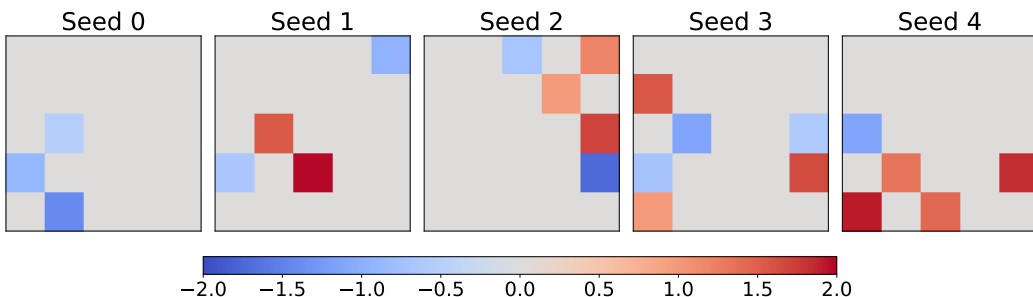

Figure 17: Ground truth weighted adjacency matrices for the experiment on the chemistry dataset.

