# OpenReview forum: "Learning Latent Structural Causal Models"
_ICLR.cc/2024/Conference — Submitted to ICLR 2024_

### Official Review · Reviewer_WQKo · 2023-10-24

**Soundness:** 1 poor
**Presentation:** 3 good
**Contribution:** 2 fair
**Rating:** 5
**Confidence:** 3

**Summary:**

This paper solves a variation of the causal representation learning problem where the joint posterior over the causal variable, the causal structure, and the parameters of the latent SCM are obtained from given high-dimensional observations.
For this purpose, the paper proposes an unsupervised approach that uses variational inference with assumptions such as linear Gaussian latent SCMs with known interventions.
The authors propose a deep learning approach to learn parameters that allow sampling of the adjacency matrix and the covariance matrix which can be used to generate samples from the training distributions.

**Strengths:**

The paper is well written. It contains a good literature review. I appreciate the authors’ effort to discuss all the necessary theoretical points that made it easy to understand their approach. Also, the experimental results are well-presented. The plots are quite useful to understand the results.

**Weaknesses:**

I provide my concerns below:

1. [Figure 2] The authors should provide more explanations about the model architecture in Figure 2. Some description in the caption would help the reader to understand the whole algorithm while reading the introduction section.

2. [Section 4.2]: The authors mentioned about obtaining q_phi(G,\theta|Z) from existing Bayesian structure learning methods. A high-level description of these methods can be provided for the reader’s convenience.

3. The ancestral sampling method should be described more explicitly.

4. [Section 5: Experiments] It is not specifically mentioned about the 3-layer neural network. For example what type of layers did the authors use? What is their dimension, and what activation functions were used? The training details can be provided in the appendix section.

Major concern:

5. [Comparison with previous work]*Although the authors discussed different recent approaches in their related work section, they did not show how their approach is different from those and how their approaches outperform earlier works. For example, the authors cited Brehmer et al. (2022) who identify the causal structure and disentangle causal variables for arbitrary, unknown causal graphs with observations before and after intervention. The assumptions in this work seem to be more relaxed than the assumptions mentioned in this paper (linear Gaussian assumption, known intervention). It is not clearly specified what improvement this paper is doing compared to previous works.

6. [Theoretical guarantee] The novelty of this paper seem to be provided in section 4.2 and 4.3 where they propose a deep learning approach to learn parameters \theta and \phi. These parameters allow us to sample the adjacency matrix and the covariance matrix. However, the authors did not discuss the identifiability of the SCM and the causal structure in detail. There is no theoretical guarantee that the algorithm will learn to sample true SCM and the causal structure. For example, the author’s one cited work Brehmer et al. (2022) claims that they can find SCMs identifiable up to a relabelling and elementwise reparameterizations of the causal variables.
Without any theoretical identifiability, what is the guarantee that the algorithm will not overfit the training datasets?  What is the guarantee that the resultant SCM and structure will match the interventional distribution (ex: distribution shift) that was absent in the training data?

7. [Baselines]: Although the authors discussed some approaches that deal with causal representation learning problems in the related work section, it is unclear why they could not find any common ground to show where they could show their comparative performance.

8. [Synthetic experiments] The authors showed their performance on a 5-node DAG with 20 random interventions. The authors should perform more intensive experiments such as: for small to large graphs with varying edge density and varying interventions.

9. [Real-world dataset] The authors used only synthetic and comparatively less complex datasets. The algorithm performance would be better observed on more complicated datasets such as Causal3DIdent [1] etc.

[1] Self-Supervised Learning with Data Augmentations Provably Isolates Content from Style.

**Questions:**

Here I provide my questions to the authors about this paper.

1. How does this algorithm fail when the SCM is not linear Gaussian additive noise SCM?
2. How is d: the number of latent variables known?
3. In a real-world setting, if Z are latent variables, how the interventions are known? How is it determined which latent variables are being intervened on?
4. The question about overfitting and finding the unique/identifiable SCM that will match unseen interventions absent in the training data.
5. How are the loss terms being calculated in Algorithm 1 lines 14,15?
 6. How \hat{L} and \hat{\sigma} are being sampled at Algorithm 1 line 3? How is gradient descent being done without breaking the computational graph since the author performs sampling at lines 3 and 10? Is the loss terms differentiable with respect to \phi and \theta even though sampling is performed?

7. What is the role of interventional datasets? How would the algorithm’s performance change when the available data is from more or less number of interventions?

8. How does the algorithm perform for different-sized DAGs with varying edge density and varying interventions?
9. What is the causal structure for the Chemistry dataset? It should be precisely mentioned.

I would request the authors to resolve my mentioned concerns and answer the questions. I am willing to increase the score if the issues are properly dealt with.

---

> ### Author Response · Authors · 2023-11-16
> **Author Rebuttal (1/n)**
>
> We would like to thank the reviewer for their detailed review and their suggestions to improve the quality of our submission.
>
> > 1. [Figure 2] The authors should provide more explanations about the model architecture in Figure 2. Some description in the caption would help the reader to understand the whole algorithm while reading the introduction section.
>
> We thank the reviewer for bringing this to our attention. In response, we have **made enhancements to both figure 2 and its accompanying description** (shown in red). We believe that these revisions contribute to an improved understanding of BIOLS in the introduction section.
>
> > 2. [Section 4.2]: The authors mentioned about obtaining $q_\phi(G,\theta|Z)$ from existing Bayesian structure learning methods. A high-level description of these methods can be provided for the reader’s convenience.
>
> While a comprehensive explanation of Bayesian structure learning is provided in an earlier section (section 3.2), we acknowledge the importance of making this connection explicit. To address this, we have introduced a line in Section 4.2 (highlighted in red), directing readers to Section 3.2. This addition clarifies the process of obtaining $q_\phi(G, \theta|Z)$. We appreciate the reviewer's attention to detail and believe this enhances the clarity of our manuscript.
>
> >3. The ancestral sampling method should be described more explicitly.
>
> Ancestral sampling is done conventionally -- topologically traversing the nodes in the graph and assigning values to each node based on the values of its parents and associated noise. To provide a more detailed explanation of ancestral sampling, we have included a footnote on page 6. We have also **added a section titled "Implementation Details" in the Appendix**, where we explicitly define ancestral sampling.
>
> > 4. [Section 5: Experiments] It is not specifically mentioned about the 3-layer neural network. For example what type of layers did the authors use? What is their dimension, and what activation functions were used? The training details can be provided in the appendix section.
>
> We thank the reviewer for bringing this to our attention. The paragraph discussing "nonlinear projection"" in Section 5 has been **revised to incorporate these details** (red text on page 8). While we utilized ReLU activation in the experiments reported in the paper, it is essential to note that, based on our observations in additional experiments, BIOLS is independent of the activation function used. It consistently performs well with alternative functions like leaky ReLU or GeLU. To ensure clarity, we have added details about the MLP, along with other related details under a section called **Implementation Details in the Appendix**.

---

> > ### Author Response · Authors · 2023-11-16
> > **Author Rebuttal (2/n)**
> >
> > > 5. [Comparison with previous work] Although the authors discussed different recent approaches in their related work section, they did not show how their approach is different from those and how their approaches outperform earlier works. For example, the authors cited Brehmer et al. (2022) who .... not clearly specified what improvement this paper is doing compared to previous works.
> >
> > We acknowledge the need to compare the properties of BIOLS with respect to prior works in the literature [1-6]. To this end, we have:
> >
> > 1. **Added a paragraph on each of these works [1-6]** to discuss their setting and assumptions in the new section titled "Situating BIOLS in the context of other related work" in the Appendix.
> >
> > 2. To concisely communicate key differences, we have **added 2 tables (Tables 1 and 2) which compare BIOLS with methods in causal discovery and causal representation learning**.
> >
> > 3. Since Brehmer et al. (2022) is the most relevant formulation, we describe here the main **advantages of BIOLS over Brehmer et al. (2022)**:
> >
> >     a. **Better scaling properties:** Figure 8 and 13 in [1] state that ILCM works for upto 8-10 causal variables. In constrast, BIOLS demonstrates scaling to atleast upto 50 nodes (added section on "Scaling the number of nodes" in the Appendix).
> >
> >    b. **Handles multi-target interventions:** ILCM [1] assumes all interventions are single-target interventions. In contrast, BIOLS supports single *and* multi-target interventions, and does not make such an assumption.
> >
> >    c. **No constraints such as paired data $(x, \tilde{x})$**: ILCM [1] requires pairs of observational and interventional data for training. This is a hard requirement, since the ELBO being optimized is a lower bound over $p(x, \tilde{x})$. In contrast, BIOLS can handle datasets that have *unequal amounts of observational ($x$) and interventional data* ($\tilde{x}$). In the extreme case, BIOLS can be used when one has *only observational or interventional data*. [1] does not address this case and ILCM cannot train on such data.
> >
> >    d. **Handles counterfactual *and* interventional data:** [1] requires that the noise remain fixed before and after an intervention which corresponds to a counterfactual setting. This is necessary for the identifiability results as well as single intervention inference in ILCM. However, this is an additional assumption. Though our paper is not about identifiability, our method (BIOLS) does not require that the noise be fixed. Regardless, for all the experiments in the main text, we keep the noise fixed for a faithful comparison with ILCM and ILCM-GT.
> >
> > ---
> > [1] Johann Brehmer, Pim De Haan, Phillip Lippe, and Taco S Cohen. Weakly supervised causal representation learning. Advances in Neural Information Processing Systems, 35:38319–38331, 2022.
> >
> > [2] Murat Kocaoglu, Christopher Snyder, Alexandros G Dimakis, and Sriram Vishwanath. Causalgan: Learning causal implicit generative models with adversarial training. In International Conference on Learning Representations, 2018.
> >
> > [3] Mengyue Yang, Furui Liu, Zhitang Chen, Xinwei Shen, Jianye Hao, and Jun Wang. Causalvae: Disentangled representation learning via neural structural causal models. In Proceedings of the IEEE/CVF Conference on Computer Vision and Pattern Recognition, pp. 9593–9602, 2021.
> >
> > [4] Xinwei Shen, Furui Liu, Hanze Dong, Qing LIAN, Zhitang Chen, and Tong Zhang. Disentangled generative causal representation learning, 2021.
> >
> > [5] Raha Moraffah, Bahman Moraffah, Mansooreh Karami, Adrienne Raglin, and Huan Liu. Causal adversarial network for learning conditional and interventional distributions, 2020.
> >
> > [6] David Lopez-Paz, Robert Nishihara, Soumith Chintala, Bernhard Scholkopf, and Leon Bottou. Discovering causal signals in images. In Proceedings of the IEEE conference on computer vision and pattern recognition, pp. 6979–6987, 2017.

---

> ### Author Response · Authors · 2023-11-16
> **Author Rebuttal (3/n)**
>
> > 6. [Theoretical guarantee] The novelty of this paper seem to be provided in section 4.2 and 4.3 where they propose a deep learning approach to learn parameters $\theta$ and $\phi$. These parameters allow us to sample the adjacency matrix and the covariance matrix. However, the authors did not discuss the identifiability of the SCM and the causal structure in detail. There is no theoretical guarantee that the algorithm will learn to sample true SCM and the causal structure.
>
> We agree that questions of identifiability are important when making conclusions about the structure of a causal model, especially for methods returning only a single structure (as in maximum likelihood methods). However in our work, we have **access to interventional data** and approximate a full **posterior distribution over the latent SCMs**, instead of returning just a single graph. Under a Bayesian treatment such as ours, questions of identifiability become less critical, as we can assign probabilities for many possible candidate graphs (and parameters) to express our level of confidence that a particular SCM yields the correct causal conclusions.
>
> **Our work provides empirical evidence for learning latent SCM in the setting of BIOLS which can motivate future works to prove identifiability in our setting.** Identifiability is an important topic and is required to know if a setting is solvable [1-4]. Works such as [4] provide good direction on proving identifiability in a closely related setup (i.e., linear Gaussian latent SCMs from low-level data). However, *our contributions are not towards identifiability* (unlike [1]). Rather, we provide empirical evidence that learning latent SCM from low-level data is possible (scaling better than [1]) and works consistently and reliably, under certain assumptions. We also show that the learnt SCM corresponds to the ground truth graph (**given by near 0 SHDs** in some of our experiments) and that **we can approximate samples from unseen interventional distributions** (figure 8). This work serves as a motivation for the authors (and possibly others in the community) to prove identifiability in the setting explored by BIOLS.
>
> **Connection between Identifiability and learnability**: Identifiability results typically state whether the causal setting is (uniquely) solvable under some assumptions. **But identifiability does not imply learnability**. As a concrete example, consider the setup of [1] where the authors prove identifiability. The algorithm ILCM introduced in [1] reliably works on only upto ~10 causal variables as reported by the authors (section 5.4, figure 8 of [1]). In contrast, BIOLS works on atleast upto 50 nodes.
>
> Furthermore, identifiability guarantees (e.g., [1]) are often true only under the infinite data sample limit. In most synthetic and real-world settings, we have only finite samples. In many problems such as in biology (finding the effect of gene knockouts), data is very limited and one might not even have sufficient finite samples. In such cases, **a Bayesian formulation such as in BIOLS can help alleviate concerns due to identifiability** by **incorporating domain-specific priors** and **providing uncertainty estimates** about the learnt latent SCM.
>
> ---
>
> [1] Johann Brehmer, Pim De Haan, Phillip Lippe, and Taco S Cohen. Weakly supervised causal representation learning. Advances in Neural Information Processing Systems, 35:38319–38331, 2022.
>
> [2] Kartik Ahuja, Jason S Hartford, and Yoshua Bengio. Weakly supervised representation learning with sparse perturbations. Advances in Neural Information Processing Systems, 35:15516–15528, 2022.
>
> [3] Kartik Ahuja, Divyat Mahajan, Yixin Wang, and Yoshua Bengio. Interventional causal representation learning. In International conference on machine learning, pp. 372–407. PMLR, 2023.
>
> [4] Liu, Yuhang, et al. "Identifying weight-variant latent causal models." arXiv preprint arXiv:2208.14153 (2022).

---

> > ### Comment · Reviewer_WQKo · 2023-11-20
> > **Importance of identifiability and suggestion about more experiments**
> >
> > I thank the authors for their effort in this paper and also for replying to each of my questions.
> >
> > I want to mention two points that I think are critical for the paper:
> >
> > 1. Identifiability: From my understanding, identifiability suggests to you what possible set of solutions you can achieve that represent the input data. In other words, given the dataset and assumptions, you can reduce your solution set to a specific family because each member of that family will correspond to the same data and assumptions. I understand, that the authors approximate a full posterior distribution over the latent SCM. This approximation will depend on the input dataset and based on that the posterior distribution will change. The proposed method can not answer which latent SCM is consistent with the input data (i.e., included in the identifiable family) and which SCM is not from the probability distribution. This is why identifiability is important.
> >
> > 2. Experiments: I would propose the authors do some experiments on well-known and more complex image datasets such as CelebA, ImageNet, or any other dataset that seems more fit to their experiment. Readers will not feel motivated to follow up on this work if the used datasets are not interesting and do not illustrate the algorithm performance clearly.

---

> > > ### Author Response · Authors · 2023-11-21
> > > **Response to Reviewer**
> > >
> > > We thank the reviewer for their comments. While we acknowledge the importance of identifiability, **we emphasize once again that identifiability in this setting merits its own work and is considered outside the scope of this work**.
> > >
> > > > 1. Identifiability suggests to you what possible set of solutions you can achieve that represent the input data.
> > >
> > > Historically in deep learning, interesting empirical discoveries have often preceded theoretical validations. **We have presented multiple experiments to demonstrate the strong performance of BIOLS**: linear projection, SO(N) rotation, nonlinear projection, as well as the chemistry dataset experiments on learning from images. All this is strong evidence that the setting is likely to be identifiable -- if it were not, we would not be able to consistently recover the latent SCM. This can inspire future work on identifiability of linear Gaussian latent SCMs similar to [2].
> > >
> > > In spite of proving identifiability, learning algorithms can struggle to recover SCMs with a large number of nodes. For example in [1], the proposed method, ILCM, does not scale beyond 8-10 nodes as noted by the authors of [1] (in section 5.4), even though the setting is identifiable. In contrast, we show BIOLS scales better with **0 SHD and near 1 AUROC scores on DAGs with upto 50 nodes**.
> > >
> > > > 2. I would propose the authors do some experiments on well-known and more complex image datasets such as CelebA, ImageNet.
> > >
> > > Regrettably, these datasets are not suitable for our setting. These datasets lack a clear ground truth graph, evaluation metrics for learned SCMs, intervention data, or a defined number of underlying nodes in the graph. Additionally, many potential causal attributes in these datasets are binary or discrete variables, differing from the continuous variables in our study. Although additional experiments and datasets might enhance the breadth of our findings, identifying suitable datasets for learning latent SCMs from pixel-level data remains an emerging area of research. This is precisely why [3] propose the chemistry dataset.
> > >
> > > > Readers will not feel motivated to follow up on this work if the used datasets are not interesting and do not illustrate the algorithm performance clearly.
> > >
> > > We respectfully disagree with the reviewer here. We are one of the first few works to propose a practical algorithm to learn latent SCM from low-level data (along with [1] and [2]), especially one that scales to 50 nodes, **as opposed to [1] and [2] which completely fail beyond 10 nodes**. As mentioned before, we have already illustrated the algorithm performance clearly (for linear and nonlinear projection, along with learning from images) and **have provided ample evidence that BIOLS can recover latent SCMs (Fig 5-14)**.
> > >
> > > Furthermore, thanks to the reviewer's feedback, we have made substantial changes to the submission which greatly improve the quality of our work. We have performed comprehensive experiments and have now **added 5 new sections** dedicated to ablation studies on graph density, interventional data, intervention value ranges, and scalability studies, complete with program runtimes. We believe that we have addressed all the reviewer's concerns and questions. Considering these additions, we kindly request the reviewer to reconsider their evaluation of our work.
> > >
> > > ---
> > > [1] Johann Brehmer, Pim De Haan, Phillip Lippe, and Taco S Cohen. Weakly supervised causal representation learning. Advances in Neural Information Processing Systems, 35:38319–38331, 2022.
> > >
> > > [2] Liu, Yuhang, et al. "Identifying weight-variant latent causal models." arXiv preprint arXiv:2208.14153 (2022).
> > >
> > > [3] Ke, Nan Rosemary, et al. "Systematic evaluation of causal discovery in visual model based reinforcement learning." arXiv preprint arXiv:2107.00848 (2021).

---

> ### Author Response · Authors · 2023-11-16
> **Author Rebuttal (4/n)**
>
> > 7. [Baselines]: Although the authors discussed some approaches that deal with causal representation learning problems in the related work section, it is unclear why they could not find any common ground to show where they could show their comparative performance.
>
> We appreciate the reviewer's insightful comments and have carefully considered their feedback. In the causal representation learning section of related works (Section 2), we primarily cite six relevant works [1-6], with [1] serving as a baseline due to its high relevance to our explored setting (refer ILCM and ILCM-GT in our submission). Below, we provide a detailed discussion on each of the referenced works, to explain why there is no common ground to include [2-5] as baselines (primarily due to unavailable code, or supervised setups that are particular to face-image datasets like CelebA):
>
> - Causal GANs [2] learn observational and interventional image distributions, specifically focusing on face-image datasets such as CelebA. This work necessitates a causal graph over **binary** image labels, as in Figure 5 of [2]. The work also assumes access to these labels and that the causal graph is already given. CausalVAE [3] also requires supervision, but is not particularly for face-image datasets. In contrast to [2] and [3], **BIOLS operates beyond face-image datasets, does not necessitate image labels, is unsupervised, and centers around SCMs with continuous variables**.
>
> - [4] addresses the training of a disentangled generative model under supervision. DEAR [3] assumes access to causal latent variables, $Z$, and the super-graph of the adjacency matrix for DAG learning. Importantly, **BIOLS, does not assume access to causal variables**, making the setting distinct. Consequently, **incorporating DEAR as a baseline would not yield a fair comparison**. Despite our efforts to evaluate DEAR by running the [official implementation](https://github.com/xwshen51/DEAR), we **encountered multiple errors in the code, suggesting potential incompleteness** (e.g., calls to undefined functions) and reproducibility issues [here](https://github.com/xwshen51/DEAR/issues/5).
>
> - CAN [5] is designed for generating images conditioned on or intervened with a set of binary labels (figure 5 in [4]), much akin to the setting of CausalGAN [2]. Notably, both works concentrate on learning face-image distributions. However, the setting of CAN is not comparable to that of BIOLS. Adding CAN as a baseline is non-trivial and at best, would introduce an apples-to-oranges comparison. **BIOLS is distinct in its focus on SCMs with continuous variables and does not involve labels, unlike CAN**. Moreover, the **code for CAN has not been provided**, and the limited rebuttal period does not allow for the timely development of code to reproduce CAN and include it in our baseline comparisons. We appreciate the reviewers' understanding of these considerations and remain committed to ensuring a fair and comprehensive evaluation of our work within the specified scope.
>
> - In reference to [6], which explores causal inference in a bivariate setting, the study proposes training a binary classifier to identify plausible causal ($X \rightarrow Y$) and anticausal ($X \leftarrow Y$) relations with the aid of labels. However, it is crucial to note the distinctions between [6] and our work, BIOLS. The setup in [6] relies on images and assumes access to bounding boxes that highlight the presence of objects in the scene, and utilizes labels for the classification task. In contrast, **BIOLS does not require bounding boxes or labels**. Furthermore, BIOLS is designed to handle structure learning over multiple nodes, demonstrated in our experiments with up to 50 nodes.
>
> ---
>
> [1] Johann Brehmer, Pim De Haan, Phillip Lippe, and Taco S Cohen. Weakly supervised causal representation learning. Advances in Neural Information Processing Systems, 35:38319–38331, 2022.
>
> [2] Murat Kocaoglu, Christopher Snyder, Alexandros G Dimakis, and Sriram Vishwanath. Causalgan: Learning causal implicit generative models with adversarial training. In International Conference on Learning Representations, 2018.
>
> [3] Mengyue Yang, Furui Liu, Zhitang Chen, Xinwei Shen, Jianye Hao, and Jun Wang. Causalvae: Disentangled representation learning via neural structural causal models. In Proceedings of the IEEE/CVF Conference on Computer Vision and Pattern Recognition, pp. 9593–9602, 2021.
>
> [4] Xinwei Shen, Furui Liu, Hanze Dong, Qing LIAN, Zhitang Chen, and Tong Zhang. Disentangled generative causal representation learning, 2021.
>
> [5] Raha Moraffah, Bahman Moraffah, Mansooreh Karami, Adrienne Raglin, and Huan Liu. Causal adversarial network for learning conditional and interventional distributions, 2020.
>
> [6] David Lopez-Paz, Robert Nishihara, Soumith Chintala, Bernhard Scholkopf, and Leon Bottou. Discovering causal signals in images. In Proceedings of the IEEE conference on computer vision and pattern recognition, pp. 6979–6987, 2017.

---

> ### Author Response · Authors · 2023-11-16
> **Author Rebuttal (5/n)**
>
> > 8. **[Synthetic experiments]** The authors showed their performance on a 5-node DAG with 20 random interventions. The authors should perform more intensive experiments such as: for small to large graphs with varying edge density and varying interventions.
>
> We appreciate the reviewer's insightful suggestion and have taken it into careful consideration. In response, we have performed the following ablation studies and summarized all these studies in the Appendix. We have updated the paper with 5 new sections in the appendix to perform comprehensive studies on BIOLS:
>
> 1. **Ablation on graph density:** We assess BIOLS's performance across ER-1, ER-2, and ER-4 graphs, characterized by $d$, $2d$, and $4d$ edges in expectation. These evaluations are conducted for both linear and nonlinear projection experiments on $d=20$ node SCMs projected to $D=100$ dimensions. Notably, we observe a trend wherein recovering edges becomes more challenging with denser graphs. This difficulty may arise from BIOLS needing to uncover a greater number of cause-effect relationships. This observation aligns with insights often noted in traditional causal discovery (e.g., Figure 5 and 12 in [1]).
>
> 2. **Ablation on number of intervention sets:** Acknowledging the reviewer's input, we extend our analysis beyond the initially reported 20 intervention sets. This ablation study involves varying the number of intervention sets, spanning from 40 to 180 sets. We conduct these experiments on ER-1 DAGs with $d=30$ and $d=50$ nodes, considering both linear and nonlinear projection scenarios. We notice a trend wherein  increasing the number of intervention sets, improves the performance of BIOLS.
>
> 3. **Ablation on range of intervention values:** During experimentation we also noticed that the range of intervention values has an impact on the performance of BIOLS. This is sensible, since a larger range of interventions provides more information about the existence (and weights) of causal connections. We perform an experiment comparing deterministic zero-valued interventions against stochastic (Gaussian with 0 mean) interventions.
>
> 4. We also perform **scaling studies** (section titled scaling the number of nodes) to study how BIOLS scales with the number of nodes (from 10 - 50 nodes). We find that BIOLS **scales atleast upto 50 nodes**.
>
> 5. **Runtimes** are reported under a separate section in the appendix (see table 5).
>
>
> > 2. How is d: the number of latent variables known?
>
> Consistent with various works in causal representation learning [2-5], we operate under the assumption that the number of causal variables is known a priori, while the specific values of the latent variables remain unknown. We emphasize that there are interesting real-world applications even under this assumption, as we note in the introduction: "*An application of interest is in the context of biology, where researchers are interested in understanding Gene Regulatory Networks (GRN). In such problems, the genes themselves are latent but can be intervened on, the results of which manifest as changes in the high-resolution images [6]. Here, **the number of latent variables (genes) is known** but the structure, mechanisms, and the image generating function remain to be uncovered.*
>
> ---
> [1] Nino Scherrer, Olexa Bilaniuk, Yashas Annadani, Anirudh Goyal, Patrick Schwab, Bernhard Schölkopf, Michael C Mozer, Yoshua Bengio, Stefan Bauer, and Nan Rosemary Ke. Learning neural causal models with active interventions. arXiv preprint arXiv:2109.02429, 2021.
>
> [2] Johann Brehmer, Pim De Haan, Phillip Lippe, and Taco S Cohen. Weakly supervised causal representation learning. Advances in Neural Information Processing Systems, 35:38319–38331, 2022.
>
> [3] Murat Kocaoglu, Christopher Snyder, Alexandros G Dimakis, and Sriram Vishwanath. Causalgan: Learning causal implicit generative models with adversarial training. In International Conference on Learning Representations, 2018.
>
> [4] Mengyue Yang, Furui Liu, Zhitang Chen, Xinwei Shen, Jianye Hao, and Jun Wang. Causalvae: Disentangled representation learning via neural structural causal models. In Proceedings of the IEEE/CVF Conference on Computer Vision and Pattern Recognition, pp. 9593–9602, 2021.
>
> [5] Xinwei Shen, Furui Liu, Hanze Dong, Qing LIAN, Zhitang Chen, and Tong Zhang. Disentangled generative causal representation learning, 2021. URL https://openreview.net/forum? id=agyFqcmgl6y.
>
> [6] Marta M Fay, Oren Kraus, Mason Victors, Lakshmanan Arumugam, Kamal Vuggumudi, John Urbanik, Kyle Hansen, Safiye Celik, Nico Cernek, Ganesh Jagannathan, et al. Rxrx3: Phenomics map of biology. bioRxiv, pp. 2023–02, 2023.

---

> ### Author Response · Authors · 2023-11-16
> **Author Rebuttal (6/n)**
>
> > 3. In a real-world setting, if Z are latent variables, how the interventions are known? How is it determined which latent variables are being intervened on?
>
> In some real-world scenarios, intervention targets are known in advance. For instance, in learning gene regulatory network structures from images reflecting gene knockouts (e.g., CRISPR), both the number of latent causal variables and the intervention targets are known [1, 2].
>
> > 5. How are the loss terms being calculated in Algorithm 1 lines 14,15?
>
> We compute the ELBO $\mathcal{L}(\phi, \theta)$ following equation (9) in the paper. In our latest revision, we explicitly state that the **observation likelihood model is Gaussian** and it decomposes as $p_\psi(\mathcal{D} \mid \mathcal{Z}, \mathcal{G}, \Theta) = p_\psi(\mathcal{D} \mid \mathcal{Z}) = \prod\limits_{i=1}^N p_\psi(\mathbf{\hat{x}_i} \mid \mathbf{\hat{z}_i})$ (refer to the red text in "Alternate factorization of the posterior" in Section 4.2).
>
> This simplifies the computation of the first term in the ELBO, $\log p_\psi(\mathcal{D} \mid \mathcal{Z})$, as it corresponds to the log likelihood of a Gaussian. The remaining terms involve computing the expected values of $\log q_\phi(L, \Sigma)$ (log Normal), $\log p(L)$ (log of a Horseshoe pdf), and $\log p(\Sigma)$ (log Normal) under the normal distribution $q_\phi(L, \Sigma)$. These operations -- sampling from a Normal, computing log Normal and log Horseshoe -- are straightforward. We emphasize that **both sampling operations in line 3 and 10 in the algorithm are reparameterized** (as clarified in response to the 6th question which appears next), ensuring the **entire forward pass is differentiable**. Consequently, obtaining gradients with respect to posterior parameters $\phi$ and likelihood parameters $\psi$ for updates is a straightforward process.
>
> > 6. How $\hat{L}$ and $\hat{\Sigma}$ are being sampled at Algorithm 1 line 3? How is gradient descent being done without breaking the computational graph since the author performs sampling at lines 3 and 10? Is the loss terms differentiable with respect to \phi and \theta even though sampling is performed?
>
> We note that in line 3, $\hat{L}$ and $\hat{\Sigma}$ is sampled from $q_\phi(L, \Sigma)$, which is a Gaussian distribution. For this, we use the well-known reparameterization trick so that gradients can flow through the sampling operation. For ancestral sampling that occurs in line 10, this is just a simple linear operation given $z_i := \widehat{W}_{*i}^T \mathbf{z} + \hat{\epsilon}_i$ and we have already alluded to this at the end of section 4.2 ("**one can perform ancestral sampling.... already reparameterized and differentiable with respect to their parameters**"). However, we make this explicit in our latest revision: As mentioned in "Ancestral sampling from $q_\phi(L, \Sigma)$" under **Implementation Details** of the appendix.
>
> $\widehat{W}_{*I}$ and $\hat{\epsilon}$ correspond to the first $K-d$ and last $d$ elements of the samples from $q\_{\phi}(L, \Sigma)$. Hence, gradients can flow to $\widehat{W}$ and $\hat{\epsilon}$ through this operation and the **entire forward pass remains differentiable with respect to $\phi$ and $\theta$**.
>
> > 7. What is the role of interventional datasets? How would the algorithm’s performance change when the available data is from more or less number of interventions?
>
> We have added experiments under a section titled **Ablation on Number of Intervention Sets** in the appendix, which we hope addresses this question. In general, increasing the number of intervention sets resulted in better performance of BIOLS (please refer figures 10 and 11), as expected.
>
> > 8. How does the algorithm perform for different-sized DAGs with varying edge density and varying interventions?
>
> Please refer to the latest revision of our paper, specifically the appendix, where we have incorporated several ablation studies for comprehensive insights. The appended sections titled **Ablation on Graph Density**, **Ablation on Number of Intervention Sets**, **Ablation on Range of Intervention Values**, and **Scaling the Number of Nodes** provide in-depth analyses that contribute to a better understanding of our proposed approach.
>
> Edge density: Figure 9 reveals that BIOLS performs better on sparser DAGs. This is expected, as the increased difficulty in receovering denser graphs is attributed to the greater number of cause-effect relationships that need to be uncovered.
>
> ---
>
> [1] Marta M Fay, Oren Kraus, Mason Victors, Lakshmanan Arumugam, Kamal Vuggumudi, John Urbanik, Kyle Hansen, Safiye Celik, Nico Cernek, Ganesh Jagannathan, et al. Rxrx3: Phenomics map of biology. bioRxiv, pp. 2023–02, 2023.
>
> [2] Srinivas Niranj Chandrasekaran, Jeanelle Ackerman, .....[author list shortened to save space]..... and Anne E. Carpenter. Jump cell painting dataset: morphological impact of 136,000 chemical and genetic perturbations. bioRxiv, 2023.

---

> ### Author Response · Authors · 2023-11-16
> **Author Rebuttal (7/n)**
>
> > 9. What is the causal structure for the Chemistry dataset? It should be precisely mentioned.
>
> The chemistry dataset [1] does not have a *single* causal structure. Rather, it allows the generation of images given a particular weighted adjacency matrix for the SCM. That is, it performs ancestral sampling to generate the latent causal variables and then uses a stochastic function to project this to pixel space. For our experiments on the chemistry dataset, we generate chemistry image datasets for 5 random ER-1 adjacency matrices generated similar to other experiments (details in "Generating the SCM" and "Generating the causal variables and intervention targets" of section 5). The images are then obtained and BIOLS is trained. That being said, we have added some visualization for the DAG structure as well as the weights of the adjacency matrix in figure 16 and 17 (in the appendix).
>
> ---
> [1] Ke, N.R., Didolkar, A., Mittal, S., Goyal, A., Lajoie, G., Bauer, S., Rezende, D., Bengio, Y., Mozer, M. and Pal, C., 2021. Systematic evaluation of causal discovery in visual model based reinforcement learning. arXiv preprint arXiv:2107.00848.

---

### Official Review · Reviewer_DxH4 · 2023-10-31

**Soundness:** 2 fair
**Presentation:** 3 good
**Contribution:** 1 poor
**Rating:** 3
**Confidence:** 4

**Summary:**

This paper introduces an empirical estimation method for inferring latent causal relationships within the framework of causal representation learning. It focuses on assuming linear latent causal models and formulates the problem as a Bayesian inference task for these models.

**Strengths:**

This paper considers the estimation of latent causal models, which is a very important task in causal representation learning.

**Weaknesses:**

The novelty of this work appears somewhat constrained. It focuses solely on the scenario where the latent causal model is linear, interventions on latent variables are assumed, and the intervention targets are known. Furthermore, it does not explicitly clarify whether this setting is theoretically identifiable.

The experimental validation is somewhat lacking. The paper only presents results with 5 latent variables, which is not enough for an empirical study.

**Questions:**

1. It is crucial for the authors to explicitly establish whether this setting is theoretically identifiable prior to introducing empirical estimation methods.

2. The paper would greatly benefit from a more comprehensive set of experiments. This should include exploring different numbers of latent variables, varying graph densities, and adjusting sample sizes for a more thorough assessment.

---

> ### Author Response · Authors · 2023-11-16
> **Author Rebuttal (1/n)**
>
> > The experimental validation is somewhat lacking. The paper only presents results with 5 latent variables, which is not enough for an empirical study.
>
> > 2. The paper would greatly benefit from a more comprehensive set of experiments. This should include exploring different numbers of latent variables, varying graph densities, and adjusting sample sizes for a more thorough assessment.
>
> We appreciate the reviewer's insightful suggestion and have taken it into careful consideration. In response, we have performed the following studies and summarized all these studies in the Appendix, in order to have the comprehensive studies required for an empirical study. Most notably, **we now have results on BIOLS on linear and nonlinear projection upto 50 nodes, with near 0 SHD and near 1 AUROC scores**. We have updated the paper with 5 new sections in the appendix to better showcase properties of BIOLS:
>
>   * **Ablation on graph density:** We assess BIOLS's performance across ER-1, ER-2, and ER-4 graphs, characterized by $d$, $2d$, and $4d$ edges in expectation. These evaluations are conducted for both linear and nonlinear projection experiments on $d=20$ node SCMs projected to $D=100$ dimensions. Notably, we observe a trend wherein recovering edges becomes more challenging with denser graphs. This difficulty may arise from BIOLS needing to uncover a greater number of cause-effect relationships. This observation aligns with insights often noted in traditional causal discovery algorithms (figure 5 and 12 in [1]).
>   * **Ablation on number of intervention sets:** Acknowledging the reviewer's input, we extend our analysis beyond the initially reported 20 intervention sets. This ablation study involves varying the number of intervention sets, spanning from 40 to 180 sets. We conduct these experiments on ER-1 DAGs with $d=30$ and $d=50$ nodes, considering both linear and nonlinear projection scenarios. We notice a trend wherein  increasing the number of intervention sets, improves the performance of BIOLS.
>   * **Ablation on range of intervention values:** During experimentation we also noticed that the range of intervention values has an impact on the performance of BIOLS. This is sensible, since a larger range of interventions provides more information about the existence (and weights) of causal connections. We perform an experiment comparing deterministic zero-valued interventions against stochastic (Gaussian with 0 mean) interventions.
>   * Additionally, we also perform **scaling studies** to study how BIOLS scales with the number of nodes, from 10 to 50 nodes in 10 node increments (please refer scaling section in the Appendix).
>   * **Runtimes** for many of our experiments are reported (across number of data samples and nodes) under a separate section in the appendix.
>
> ---
> [1] Nino Scherrer, Olexa Bilaniuk, Yashas Annadani, Anirudh Goyal, Patrick Schwab, Bernhard Schölkopf, Michael C Mozer, Yoshua Bengio, Stefan Bauer, and Nan Rosemary Ke. Learning neural causal models with active interventions. arXiv preprint arXiv:2109.02429, 2021.

---

> ### Author Response · Authors · 2023-11-16
> **Author Rebuttal (2/n)**
>
> > Furthermore, it does not explicitly clarify whether this setting is theoretically identifiable.
>
> We agree that questions of identifiability are important when making conclusions about the structure of a causal model, especially for methods returning only a single structure (as in maximum likelihood methods). However in our work, we have **access to interventional data** and approximate a full **posterior distribution over the latent SCMs**, instead of returning just a single graph. Under a Bayesian treatment such as ours, questions of identifiability become less critical, as we can assign probabilities for many possible candidate graphs (and parameters) to express our level of confidence that a particular SCM yields the correct causal conclusions.
>
> **Our work provides empirical evidence for learning latent SCM in the setting of BIOLS which can motivate future works to prove identifiability in our setting.** Identifiability is an important topic and is required to know if a setting is solvable [1-4]. Works such as [4] provide good direction on proving identifiability in a closely related setup (i.e., linear Gaussian latent SCMs from low-level data). However, *our contributions are not towards identifiability* (unlike [1]). Rather, we provide empirical evidence that learning latent SCM from low-level data is possible (scaling better than [1]) and works consistently and reliably, under certain assumptions. We also show that the learnt SCM corresponds to the ground truth graph (**given by near 0 SHDs** in some of our experiments) and that **we can approximate samples from unseen interventional distributions** (figure 8). This work serves as a motivation for the authors (and possibly others in the community) to study identifiability in the setting explored by BIOLS.
>
> **Connection between Identifiability and learnability**: Identifiability results typically state whether the causal setting is (uniquely) solvable under some assumptions. **But identifiability does not imply learnability**. As a concrete example, consider the setup of [1] where the authors prove identifiability. The algorithm ILCM introduced in [1] reliably works on only upto ~10 causal variables as reported by the authors (section 5.4, figure 8 of [1]). In contrast, BIOLS works on atleast upto 50 nodes.
>
> These identifiability guarantees (e.g., [1]) are often true only under the infinite data sample limit. In most synthetic and real-world settings, we have only finite samples. In many problems such as in biology (finding the effect of gene knockouts), data is very limited and one might not even have sufficient finite samples. In such cases, **a Bayesian formulation such as in BIOLS can help alleviate concerns due to identifiability** by **incorporating domain-specific priors** and **providing uncertainty estimates** about the learnt latent SCM.
>
> ---
> [1] Johann Brehmer, Pim De Haan, Phillip Lippe, and Taco S Cohen. Weakly supervised causal representation learning. Advances in Neural Information Processing Systems, 35:38319–38331, 2022.
>
> [2] Kartik Ahuja, Jason S Hartford, and Yoshua Bengio. Weakly supervised representation learning with sparse perturbations. Advances in Neural Information Processing Systems, 35:15516–15528, 2022.
>
> [3] Kartik Ahuja, Divyat Mahajan, Yixin Wang, and Yoshua Bengio. Interventional causal representation learning. In International conference on machine learning, pp. 372–407. PMLR, 2023.
>
> [4] Liu, Yuhang, et al. "Identifying weight-variant latent causal models." arXiv preprint arXiv:2208.14153 (2022).

---

### Official Review · Reviewer_YGzo · 2023-10-31

**Soundness:** 2 fair
**Presentation:** 3 good
**Contribution:** 2 fair
**Rating:** 3
**Confidence:** 4

**Summary:**

This paper investigates the learning of latent causal structures from low-level observational data with known interventions. The authors primarily concentrate on learning a linear latent causal model and employ Bayesian inference methods to tackle this learning task. Additionally, they conducted experiments using synthetic and the image dataset to validate the effectiveness of their proposed approach.

**Strengths:**

This paper is well written with clear motivation.

What the authors focused on is indeed an interesting yet challenging research topic in causal inference and machine learning.

**Weaknesses:**

Novelty: In my opinion, the authors introduced an approach for parameter estimation through deep learning methods. However, it's worth noting that they didn't provide a theoretical analysis to support their approach. That is to say, the authors did not offer an analysis of the identifiability of the latent causal model. Without theoretical identifiability results, it becomes challenging to have full confidence in the outcomes generated by their proposed method.

Experiments: The experimental results only demonstrated a basic setting with five nodes, which may not be sufficient to provide a comprehensive empirical study.

**Questions:**

Regarding the number of latent variables:  How can we get the number of latent variables? Do we need to know it in advance?

Regarding the intervention: Is the intervening variable only on the observed variable? Can we intervet the latent variables?

Regarding the experiments: What is the performance of different setting graphs?

---

> ### Author Response · Authors · 2023-11-16
> **Author Rebuttal (1/n)**
>
> > Experiments: The experimental results only demonstrated a basic setting with five nodes, which may not be sufficient to provide a comprehensive empirical study.
>
> We appreciate the reviewer's insightful suggestion and have taken it into careful consideration. In response, we have performed the following ablation studies  and summarized all these studies in the Appendix. The updated version of the paper contains:
>
>   * **Ablation on number of intervention sets:** Acknowledging the reviewer's input, we extend our analysis beyond the initially reported 20 intervention sets. This ablation study involves varying the number of intervention sets, spanning from 40 to 180 sets. We conduct these experiments on ER-1 DAGs with $d=30$ and $d=50$ nodes, considering both linear and nonlinear projection scenarios. We notice a trend wherein  increasing the number of intervention sets, improves the performance of BIOLS.
>   * **Ablation on range of intervention values:** During experimentation we also noticed that the range of intervention values has an impact on the performance of BIOLS. This is sensible, since a larger range of interventions provides more information about the existence (and weights) of causal connections. We perform an experiment comparing deterministic zero-valued interventions against stochastic (Gaussian with 0 mean) interventions.
>   * Additionally, we also perform **scaling studies** (section titled scaling the number of nodes) to study how BIOLS scales with the number of nodes (from 10 - 50 nodes). We find that BIOLS scales **atleast upto 50 nodes**.
>   * **Runtimes** for many of our experiments are reported (across number of data samples and nodes) under a separate section in the appendix.
>
>
> > Regarding the number of latent variables: How can we get the number of latent variables? Do we need to know it in advance?
>
> Consistent with various works in causal representation learning literature[1-4], we operate under the standard assumption that the number of causal variables is known. We emphasize that there are interesting real-world applications even under this assumption, as we note in the introduction: "*An application of interest is in the context of biology, where researchers are interested in understanding Gene Regulatory Networks (GRN). In such problems, the genes themselves are latent but can be intervened on, the results of which manifest as changes in the high-resolution images [5]. Here, **the number of latent variables (genes) is known** but the structure, mechanisms, and the image generating function remain to be uncovered.*"
>
> > Regarding the intervention: Is the intervening variable only on the observed variable? Can we intervet the latent variables?
>
> All interventions discussed in the paper are on latent variables, *not on the observed variables*.
>
> > Regarding the experiments: What is the performance of different setting graphs?
>
> By "different setting graphs", does the reviewer mean graph of varying densities? If so, we have performed an ablation and have added it in our latest revision of the paper (please refer to section on "**Ablation on graph density**" in the Appendix). We assess BIOLS's performance across ER-1, ER-2, and ER-4 graphs, characterized by $d$, $2d$, and $4d$ edges in expectation. These evaluations are conducted for both linear and nonlinear projection experiments on $d=20$ node SCMs projected to $D=100$ dimensions. Notably, we observe a trend wherein recovering edges becomes more challenging with denser graphs. This difficulty arises from BIOLS needing to uncover a greater number of cause-effect relationships. Does this address the reviewer's concern?
>
> ---
>  [1] Johann Brehmer, Pim De Haan, Phillip Lippe, and Taco S Cohen. Weakly supervised causal representation learning. Advances in Neural Information Processing Systems, 35:38319–38331, 2022.
>
> [2] Murat Kocaoglu, Christopher Snyder, Alexandros G Dimakis, and Sriram Vishwanath. Causalgan: Learning causal implicit generative models with adversarial training. In International Conference on Learning Representations, 2018.
>
> [3] Mengyue Yang, Furui Liu, Zhitang Chen, Xinwei Shen, Jianye Hao, and Jun Wang. Causalvae: Disentangled representation learning via neural structural causal models. In Proceedings of the IEEE/CVF Conference on Computer Vision and Pattern Recognition, pp. 9593–9602, 2021.
>
> [4] Xinwei Shen, Furui Liu, Hanze Dong, Qing LIAN, Zhitang Chen, and Tong Zhang. Disentangled generative causal representation learning, 2021.
> [5] Marta M Fay, Oren Kraus, Mason Victors, Lakshmanan Arumugam, Kamal Vuggumudi, John Urbanik, Kyle Hansen, Safiye Celik, Nico Cernek, Ganesh Jagannathan, et al. Rxrx3: Phenomics map of biology. bioRxiv, pp. 2023–02, 2023.

---

> ### Author Response · Authors · 2023-11-16
> **Author Rebuttal (2/n)**
>
> > Novelty: In my opinion, the authors introduced an approach for parameter estimation through deep learning methods. However, it's worth noting that they didn't provide a theoretical analysis to support their approach. That is to say, the authors did not offer an analysis of the identifiability of the latent causal model. Without theoretical identifiability results, it becomes challenging to have full confidence in the outcomes generated by their proposed method.
>
> **Our work provides empirical evidence for learning latent SCM in the setting of BIOLS which can motivate future works to prove identifiability in our setting.** Identifiability is an important topic and is required to know if a setting is solvable [1-4]. Works such as [4] provide good direction on proving identifiability in a closely related setup (i.e., linear Gaussian latent SCMs from low-level data). However, *our contributions are not towards identifiability* (unlike [1]). Rather, we provide empirical evidence that learning latent SCM from low-level data is possible (scaling better than [1]) and works consistently and reliably, under certain assumptions. We also show that the learnt SCM corresponds to the ground truth graph (**given by near 0 SHDs** in some of our experiments) and that **we can approximate samples from unseen interventional distributions** (figure 8). This work serves as a motivation for the authors (and possibly others in the community) to study identifiability in the setting explored by BIOLS.
>
> We agree that questions of identifiability are important when making conclusions about the structure of a causal model, especially for methods returning only a single structure (as in maximum likelihood methods). However in our work, we have **access to interventional data** and approximate a full **posterior distribution over the latent SCMs**, instead of returning just a single graph. Under a Bayesian treatment such as ours, questions of identifiability become less critical, as we can assign probabilities for many possible candidate graphs (and parameters) to express our level of confidence that a particular SCM yields the correct causal conclusions.
>
> **Connection between Identifiability and learnability**: Identifiability results typically state whether the causal setting is (uniquely) solvable under some assumptions. **But identifiability does not imply learnability**. As a concrete example, consider the setup of [1] where the authors prove identifiability. The algorithm ILCM introduced in [1] reliably works on only upto ~10 causal variables as reported by the authors (section 5.4, figure 8 of [1]). In contrast, BIOLS works on atleast upto 50 nodes.
>
> These identifiability guarantees (e.g., [1]) are often true only under the infinite data sample limit. In most synthetic and real-world settings, we have only finite samples. In many problems such as in biology (finding the effect of gene knockouts), data is very limited and one might not even have sufficient finite samples. In such cases, **a Bayesian formulation such as in BIOLS can help alleviate concerns due to identifiability** by **incorporating domain-specific priors** and **providing uncertainty estimates** about the learnt latent SCM.
>
> ---
> [1] Johann Brehmer, Pim De Haan, Phillip Lippe, and Taco S Cohen. Weakly supervised causal representation learning. Advances in Neural Information Processing Systems, 35:38319–38331, 2022.
>
> [2] Kartik Ahuja, Jason S Hartford, and Yoshua Bengio. Weakly supervised representation learning with sparse perturbations. Advances in Neural Information Processing Systems, 35:15516–15528, 2022.
>
> [3] Kartik Ahuja, Divyat Mahajan, Yixin Wang, and Yoshua Bengio. Interventional causal representation learning. In International conference on machine learning, pp. 372–407. PMLR, 2023.
>
> [4] Liu, Yuhang, et al. "Identifying weight-variant latent causal models." arXiv preprint arXiv:2208.14153 (2022).
>
> ---
> ---
>
> We sincerely appreciate the thoughtful comments from the reviewer, which undoubtedly enhance the quality of our work. We believe our responses address the concerns raised. If these clarifications align with the reviewer's expectations, we kindly request a reconsideration of the score. However, if there are any remaining concerns, we are eager to address them promptly.

---

> > ### Comment · Reviewer_YGzo · 2023-12-05
> > **Response**
> >
> > Thank you for the authors' efforts. Having carefully reviewed the response and the comments from other reviewers, I have decided to maintain my current score.

---

### Official Review · Reviewer_h7nM · 2023-11-01

**Soundness:** 2 fair
**Presentation:** 2 fair
**Contribution:** 2 fair
**Rating:** 5
**Confidence:** 3

**Summary:**

This work seeks to present a practical approach for inferring latent linear-Gaussian causal models solely from observations, utilizing a Bayesian framework. The method entails breaking down the process of inferring latent causal variables into two main components: inferring the latent weight matrix and estimating latent Gaussian noises. To validate the effectiveness of the proposed method, it conduct experiments on synthetic datasets and an image dataset.

**Strengths:**

My primary concerns are as follows:

1) Theoretical guarantees: It is widely recognized that identifying latent causal models is a challenging task without the incorporation of additional assumptions. Recent studies have made significant progress in demonstrating the identifiability of latent causal models by exploring the change of weights, such as hard and soft interventions [1][2], on the model's weights. Nevertheless, there has been a noticeable absence of discussion regarding how our proposed method satisfies the assumptions necessary for achieving these identifiability results.

2) Contributions: The primary contribution of this work lies in the development of a practical method for inferring latent causal models. However, from a technical perspective, the contributions are somewhat limited, as the technical details closely resemble those of previous work [3], even though this study uncovers causal models in a latent space. Additionally, as a practical method, the experiments conducted in this work are somewhat lacking in comprehensiveness. For instance, the image dataset utilized in this study is relatively simple, which may not sufficiently validate the advantages of the proposed method. To enhance the robustness of the findings, I would suggest the author consider using more complex datasets, such as Causal3DIdent in [4] and CausalCircuit in [1]. Furthermore, it is imperative to perform a comparative analysis of the proposed methods against existing approaches, such as those in [1].

3) Several critical details remain unaddressed, such as the proposed method to ensure that the learned causal models conform to a Directed Acyclic Graph (DAG) structure.


[1] Brehmer, Johann, et al. "Weakly supervised causal representation learning." Advances in Neural Information Processing Systems 35 (2022): 38319-38331.
[2] Liu, Yuhang, et al. "Identifying weight-variant latent causal models." arXiv preprint arXiv:2208.14153 (2022).
[3] Cundy, Chris, Aditya Grover, and Stefano Ermon. "Bcd nets: Scalable variational approaches for bayesian causal discovery." Advances in Neural Information Processing Systems 34 (2021): 7095-7110.
[4] Von Kügelgen, Julius, et al. "Self-supervised learning with data augmentations provably isolates content from style." Advances in neural information processing systems 34 (2021): 16451-16467.

**Weaknesses:**

See above

**Questions:**

See above

---

> ### Author Response · Authors · 2023-11-17
> **Author Rebuttal (1/n)**
>
> > 1. Theoretical guarantees: It is widely recognized that identifying latent causal models is a challenging task without the incorporation of additional assumptions. Recent studies have made significant progress in demonstrating the identifiability of latent causal models by exploring the change of weights, such as hard and soft interventions [1][2], on the model's weights. Nevertheless, there has been a noticeable absence of discussion regarding how our proposed method satisfies the assumptions necessary for achieving these identifiability results.
>
> **Our work provides empirical evidence for learning latent SCM in the setting of BIOLS which can motivate future works to prove identifiability in our setting.** Identifiability is an important topic and is required to know if a setting is solvable [1-4]. Works such as [4] provide good direction on proving identifiability in a closely related setup (i.e., linear Gaussian latent SCMs from low-level data). However, *our contributions are not towards identifiability* (unlike [1]). Rather, we provide empirical evidence that learning latent SCM from low-level data is possible (scaling better than [1]) and works consistently and reliably, under certain assumptions. We also show that the learnt SCM corresponds to the ground truth graph (**given by near 0 SHDs** in some of our experiments) and that **we can approximate samples from unseen interventional distributions** (figure 8). This work serves as a motivation for the authors (and possibly others in the community) to study identifiability in the setting explored by BIOLS.
>
> We agree that questions of identifiability are important when making conclusions about the structure of a causal model, especially for methods returning only a single structure (as in maximum likelihood methods). However in our work, we have **access to interventional data** and approximate a full **posterior distribution over the latent SCMs**, instead of returning just a single graph. Under a Bayesian treatment such as ours, questions of identifiability become less critical, as we can assign probabilities for many possible candidate graphs (and parameters) to express our level of confidence that a particular SCM yields the correct causal conclusions.
>
> **Connection between Identifiability and learnability**: Identifiability results typically state whether the causal setting is (uniquely) solvable under some assumptions. **But identifiability does not imply learnability**. As a concrete example, consider the setup of [1] where the authors prove identifiability. The algorithm ILCM introduced in [1] reliably works on only upto ~10 causal variables as reported by the authors (section 5.4, figure 8 of [1]). **In contrast, BIOLS works on atleast upto 50 nodes** (Figure 15 of our submission, section "Scaling the number of nodes" in the Appendix), **obtaining near 0 SHDs**.
>
> These identifiability guarantees (e.g., [1]) are often true only under the infinite data sample limit. In most synthetic and real-world settings, we have only finite samples. In many problems such as in biology (finding the effect of gene knockouts), data is very limited and one might not even have sufficient finite samples. In such cases, **a Bayesian formulation such as in BIOLS can help alleviate concerns due to identifiability** by **incorporating domain-specific priors** and **providing uncertainty estimates** about the learnt latent SCM.
>
> ---
> [1] Johann Brehmer, Pim De Haan, Phillip Lippe, and Taco S Cohen. Weakly supervised causal representation learning. Advances in Neural Information Processing Systems, 35:38319–38331, 2022.
>
> [2] Kartik Ahuja, Jason S Hartford, and Yoshua Bengio. Weakly supervised representation learning with sparse perturbations. Advances in Neural Information Processing Systems, 35:15516–15528, 2022.
>
> [3] Kartik Ahuja, Divyat Mahajan, Yixin Wang, and Yoshua Bengio. Interventional causal representation learning. In International conference on machine learning, pp. 372–407. PMLR, 2023.
>
> [4] Liu, Yuhang, et al. "Identifying weight-variant latent causal models." arXiv preprint arXiv:2208.14153 (2022).

---

> ### Author Response · Authors · 2023-11-17
> **Author Rebuttal (2/n)**
>
> > Additionally, as a practical method, the experiments conducted in this work are somewhat lacking in comprehensiveness.
>
> We appreciate the reviewer's insightful suggestion and have taken it into careful consideration. In response, we have performed the following studies and summarized all these studies in the Appendix, in order to have the comprehensive studies required for an empirical study. Most notably, **we now have results on BIOLS on linear and nonlinear projection upto 50 nodes, with near 0 SHD and near 1 AUROC scores**. We have updated the paper with 5 new sections in the appendix to better showcase properties of BIOLS:
>
> * **Ablation on graph density:** We assess BIOLS's performance across ER-1, ER-2, and ER-4 graphs, characterized by $d$, $2d$, and $4d$ edges in expectation. These evaluations are conducted for both linear and nonlinear projection experiments on $d=20$ node SCMs projected to $D=100$ dimensions. Notably, we observe a trend wherein recovering edges becomes more challenging with denser graphs. This difficulty may arise from BIOLS needing to uncover a greater number of cause-effect relationships. This observation aligns with insights often noted in traditional causal discovery algorithms (e.g., Figure 5 and 12 in [2]).
>   * **Ablation on number of intervention sets:** Acknowledging the reviewer's input, we extend our analysis beyond the initially reported 20 intervention sets. This ablation study involves varying the number of intervention sets, spanning from 40 to 180 sets. We conduct these experiments on ER-1 DAGs with $d=30$ and $d=50$ nodes, considering both linear and nonlinear projection scenarios. We notice a trend wherein  increasing the number of intervention sets, improves the performance of BIOLS.
>   * **Ablation on number of intervention sets:** During experimentation we also noticed that the range of intervention values has an impact on the performance of BIOLS. This is sensible, since a larger range of interventions provides more information about the existence (and weights) of causal connections. We perform an experiment comparing deterministic zero-valued interventions against stochastic (Gaussian with 0 mean) interventions.
>   * Additionally, we also perform **scaling studies** to study how BIOLS scales with the number of nodes, from 10 to 50 nodes in 10 node increments (please refer scaling section in the Appendix).
>   * **Runtimes** for many of our experiments are reported (across number of data samples and nodes) under a separate section in the appendix.
>
> > Furthermore, it is imperative to perform a comparative analysis of the proposed methods against existing approaches, such as those in [1].
>
> We completely agree with the reviewer here, and would like to **emphasize that the initial submission already contains experiments that compare BIOLS to [1]**. This has been explicitly mentioned under "Baselines" in section 5 of the main text. We consider two variants: ILCM (as proposed in [1]) and a modified version called ILCM-GT where intervention targets are given to the model, just like BIOLS, for a fair comparison. Experiments in our submission already show that BIOLS outperforms BIOLS on ER-1 and ER-2 DAGs where the projection function is an SO(N) rotation (Figure 5, row 1), a linear projection (Figure 5, row 2), or a nonlinear projection (figure 6). We believe that this addresses the reviewers concern. Additionally, we qualitatively compare BIOLS with other related work in causal representation learning (Appendix, section "**Situating BIOLS in the context of other related work**").
>
> Moreover, the algorithm introduced in [1] reliably works on only upto ~10 causal variables as acknowledged by the authors of [1] (section 5.4, figure 8 of [1]). In contrast, BIOLS works on atleast upto 50 nodes (section on "scaling number of nodes" in Appendix) with near 0 SHDs.
>
> ---
>
> [1] Johann Brehmer, Pim De Haan, Phillip Lippe, and Taco S Cohen. Weakly supervised causal representation learning. Advances in Neural Information Processing Systems, 35:38319–38331, 2022.
>
> [2] Nino Scherrer, Olexa Bilaniuk, Yashas Annadani, Anirudh Goyal, Patrick Schwab, Bernhard Schölkopf, Michael C Mozer, Yoshua Bengio, Stefan Bauer, and Nan Rosemary Ke. Learning neural causal models with active interventions. arXiv preprint arXiv:2109.02429, 2021.

---

> ### Author Response · Authors · 2023-11-17
> **Author Rebuttal (3/n)**
>
> > 3. Several critical details remain unaddressed, such as the proposed method to ensure that the learned causal models conform to a Directed Acyclic Graph (DAG) structure.
>
> We thank the reviewer for pointing this out. In response, we have **included a dedicated section titled "Implementation Details"** in the Appendix of our revised manuscript. In this section, we intricately describe the conventional ancestral sampling process, crucial to our methodology. Specifically, we illustrate the initial step of drawing samples from $q_\phi(L, \Sigma)$, where $L$ contains $d(d-1)/2$ elements, representing the lower triangle of the $d \times d$ adjacency matrix. This deliberate parameterization is crucial in enforcing the acyclic nature of the DAGs. Notably, an adjacency matrix featuring a fully populated lower triangle would imply a fully connected DAG. In this configuration, node $j$ would encompass nodes $1 \dots (j-1)$ as its parent variables. The significance of this parameterization becomes apparent in our ability to perform ancestral sampling, preventing the emergence of cyclic dependencies. By exclusively parameterizing the lower triangle, we enable the encompassment of all potential DAGs for a given node ordering. This ensures that the learned weighted adjacency in BIOLS is *always a DAG*.

---

> ### Comment · Reviewer_h7nM · 2023-11-22
> **Comments**
>
> Thank you for such detailed clarification.
>
> I agree the significance of bridging the gap between identifiability and empirical methods, encompassing a thorough understanding of this specific gap, and the development of effective methods. I commend the authors for their efforts on this topic. Unfortunately, I think that the current work, when viewed through the lens of empirical methods, does not quite ignite my excitement. Consequently, my rating is limited to 5.
>
> I encorage the authors to consider the following questions using empirical methods:
>
> 1) How does sample size/ the number of latent variables influence the performance of recovering latent variables and graph structures, and insights?  While I observed that the authors conducted experiments on performance concerning the number of latent variables, a more in-depth analysis appears to be warranted.
>
> 2) How does the reconstruction error (on complicated dataset, e.g., Causal3DIdent and CausalCircuit) impact the effectiveness of recovering latent variables and graph structures, and subsequently, the insights gained? While existing works on identifiability often recover latent causal variables by matching marginal data distributions, the specific influence of reconstruction error in this context remains unclear.
>
> 3) How does the presence of a small error between the recovered variables and the true ones influence the learned graph structure, and what are the underlying reasons for this impact? Empirically, it's common to observe a small error between the recovered variables and the true ones. Investigating the implications of this error on the learned graph structure can provide valuable insights into the robustness and reliability of the recovery process.
>
> 4) How does the scaling indeterminacy influence the performance of recovering latent variables and graph structures, and any suggestions? The scaling indeterminacy often arises due to the inherent ambiguity in the scaling of latent variables.
>
> ...

---

### Meta-Review · Area_Chair_DVRg · 2023-12-04

**Metareview:**

This paper proposed a method named BIOLS for learning latent causal structures from low-level data, which is an approximate inference method that performs joint inference over the causal variables, structure, and parameters of the latent SCM from known interventions.

A reasonable amount of discussions took place between the authors and the reviewers. In the end, we got four reviews with ratings of 5, 3, 3, and 5 with confidence of 3, 4, 4, and 3 respectively.

All the reviewers are concerned about the theoretical guarantees, including the identifiability of this method, and the authors would need to address the raised critical concerns related to the theoretical aspects. The decision is reject.

**Justification For Why Not Higher Score:**

The identifiability of this method is the main concern and the authors need to address such concerns properly.

**Justification For Why Not Lower Score:**

N/A

---

### Decision · Program_Chairs · 2024-01-16

Reject